# Train Once and Explain Everywhere: Pre-training Interpretable Graph Neural Networks

**Jun Yin**[*]
Central South University
yinjun2000@csu.edu.cn

**Chaozhuo Li**[*]
Microsoft Research Asia
cli@microsoft.com

**Hao Yan**
Central South University
CSUyh1999@csu.edu.cn

**Jianxun Lian**
Microsoft Research Asia
jianxun.lian@microsoft.com

**Senzhang Wang**[†]
Central South University
szwang@csu.edu.cn

## Abstract

Intrinsic interpretable graph neural networks aim to provide transparent predictions by identifying the influential fraction of the input graph that guides the model prediction, i.e., the explanatory subgraph. However, current interpretable GNNs mostly are dataset-specific and hard to generalize to different graphs. A more generalizable GNN interpretation model which can effectively distill the universal structural patterns of different graphs is until-now unexplored. Motivated by the great success of recent pre-training techniques, we for the first time propose the **P**re-training **I**nterpretable **G**raph **N**eural **N**etwork ($\pi$-**GNN**[3]) to distill the universal interpretability of GNNs by pre-training over synthetic graphs with ground-truth explanations. Specifically, we introduce a structural pattern learning module to extract diverse universal structure patterns and integrate them together to comprehensively represent the graphs of different types. Next, a hypergraph refining module is proposed to identify the explanatory subgraph by incorporating the universal structure patterns with local edge interactions. Finally, the task-specific predictor is cascaded with the pre-trained $\pi$-GNN model and fine-tuned over downstream tasks. Extensive experiments demonstrate that $\pi$-GNN significantly surpasses the leading interpretable GNN baselines with up to 9.98% interpretation improvement and 16.06% classification accuracy improvement. Meanwhile, $\pi$-GNN pre-trained on graph classification task also achieves the top-tier interpretation performance on node classification task, which further verifies its promising generalization performance among different downstream tasks.

## 1 Introduction

Although graph neural networks (GNNs) [1, 2, 3, 4, 5] have achieved remarkable success in various applications [6, 7, 8, 9], their black-box nature prevents humans from understanding the inner decision-making mechanism [10, 11]. This issue calls for the development of intrinsic interpretable GNNs [12, 13, 14], which can reveal the mystery of *"Which fraction of the input graph is the most vital and leads to the model prediction?"* Intrinsic interpretable GNNs aim to identify an influential subgraph of the input graph, i.e., the explanation, and make final predictions under the guidance of the explanatory subgraph [12, 13, 14]. Constructing interpretable GNNs makes it possible to investigate

---

[*]Equal contribution.
[†]Corresponding author.
[3]Our code and datasets are available at https://anonymous.4open.science/r/PI-GNN-F86C

37th Conference on Neural Information Processing Systems (NeurIPS 2023).

the decision-making mechanism and justify model predictions, which is critically important to develop trustworthy artificial intelligence.

However, a major issue of the existing interpretable GNNs [12, 13, 14] is that they are mostly dataset-specific, which means an interpretable GNN model trained on a certain graph dataset (e.g., the semantic networks [15, 16, 17]) usually does not work well on another graph dataset (e.g., the molecular graphs [18]). It is also difficult for an interpretable GNN trained on a certain task (e.g., graph classification [7]) to generalize to another task (e.g., node classification [6]). Compared to the text or image data [19, 20], graphs with non-Eucildean structure are often associated with substantial node features of various domains, making the pre-training method in NLP and CV hard to be directly applied [21]. Recently graph representation learning based pre-training methods have been investigated [22, 21, 23, 24], however, they target at downstream prediction tasks but ignore the interpretability of the GNNs. It is widely acknowledged that the topological structure of various graphs generally follows some universal structural patterns or properties which are transferable [22], such as the scale-free property [25], the motif distribution [26], and the core-periphery structure [27]. Therefore, in this paper we argue that the intrinsic GNN interpretation contains some universal structural patterns, which are independent of the downstream tasks and generalizable to different types of graphs. Motivated by the great success of pre-training technique [19, 20, 22, 21], we for the first time study: *whether we can and how to construct a pre-training interpretable GNN that is general enough to work well on different types of graphs and downstream tasks?*

The challenges of designing a pre-training interpretable GNN are three-fold. First, labeling ground-truth explanation in the real-world graphs is extremely resource- and time-comsuming [28]. The lacking of ground-truth explanation makes it hard to distill the universal interpretability in the pre-training phase. Second, multiple structural patterns usually co-exist in one graph dataset, such as the scale-free pattern and the motif distribution pattern in chemical molecule graphs [29]. How to extract and integrate multiple structural patterns for a more comprehensive and general graph representation during pre-training is also challenging. Third, the local structural interactions such as the neighbor edges interaction [30] should be considered when identifying explanations. However, due to the structural diversity of different local neighborhoods, it is non-trivial to incorporate the global structural patterns with the local structural interactions.

In this papaer, we propose a **P**re-training **I**nterpretable **G**raph **N**eural **N**etwork model ($\pi$-**GNN** for short), which is first pre-trained over a large synthetic graph dataset with ground-truth explanations and then fine-tuned on different downstream datasets and tasks. Specifically, we first construct a synthetic graph dataset named PT-Motifs that contains various structural patterns and ground truth explanations, over which the $\pi$-GNN model is pre-trained. Considering the co-existence of multiple structural patterns, a structural pattern learning module is introduced to extract and integrate multiple structural patterns to make them generalizable to diverse graph datasets. To better identify the explanatory subgraph, a hypergraph refining module is also proposed to capture the local structural interaction and incorporate it with the universal structural patterns. It is also proved the structural representation ability of $\pi$-GNN can approach the theoretical upper bound through the hypergraph refining process. Our main contributions are summarized as follows.

- We for the first time propose a pre-training interpretable GNN model $\pi$-GNN, which can be generalized to different graph datasets and diverse downstream tasks.

- To extract the universal interpretability of $\pi$-GNN, we construct a synthetic graph classification dataset PT-Motifs with ground-truth explanations for pre-training.

- Two innovative modules, i.e., the structural pattern learning module and the hypergraph refining module, are designed and integrated into $\pi$-GNN. The former captures and integrates multiple universal structural patterns for generalizable graph representation. The latter incorporates the universal patterns with local structural interactions to identify the explanation.

- Compared with the SOTA baselines, $\pi$-GNN achieves up to 9.98% ROC-AUC improvement in interpretation and 16.06% accuracy improvement in prediction. Moreover, $\pi$-GNN pre-trained on graph classification dataset is able to achieve comparable performance with the leading baselines on the node classification task.

## 2 Background

In this section, we briefly introduce the graph neural networks and the GNN explanation methods[4]. The key notations are summarized in Appendix A for clarity.

**Graph Neural Networks.** Graph structure data can be denoted as $G = (\mathcal{V}, \mathcal{E})$ with the node set $\mathcal{V}$ and the edge set $\mathcal{E}$. The node features are represented as the matrix $\mathbf{X} \in \mathbb{R}^{|\mathcal{V}| \times d}$ and the edge features (if exist) are represented as $\mathbf{X}_E \in \mathbb{R}^{|\mathcal{E}| \times d_E}$. The topological structure is usually represented as an adjacency matrix $\mathbf{A} \in \mathbb{R}^{|\mathcal{V}| \times |\mathcal{V}|}$, where the element $A_{ij} = 1$ indicates the edge $(i, j)$ exists and $A_{ij} = 0$ otherwise. Graph neural networks (GNNs) aim to learn expressive representation on graphs for the downstream tasks [1, 2, 3, 4, 32, 33], such as graph classification [7], node classification [6, 34], and link prediction [35, 36]. Typically, to learn the representation of node $v_i$, GNNs aggregate the information from its neighborhood $\mathcal{N}(v_i)$ and then combine it with $v_i$'s own features. For example, the operation of the $k$-th GCNs layer can be formulated as follows [1],

$$\mathbf{X_{k+1}} = F\Big(\mathbf{D}^{-\frac{1}{2}}\hat{\mathbf{A}}\mathbf{D}^{-\frac{1}{2}}\mathbf{X_k}\mathbf{W_k}\Big), \tag{1}$$

where $\mathbf{X_k}$ and $\mathbf{X_{k+1}}$ are the input and output of the $k$-th layer, $\hat{\mathbf{A}} = \mathbf{A} + \mathbf{I}$ is the adjacent matrix with self-loops, and $\mathbf{D}$ is a diagonal matrix whose element $D_{i,i}$ represents the degree of $v_i$. $\mathbf{W_k}$ is a trainable matrix and $F(\cdot)$ is a non-linear activation function.

**GNN Explanation Methods.** GNN explanation methods can be catogrized into the post-hoc explanation methods [10, 11, 37, 38, 39] and the intrinsic interpretable methods [3, 40, 12, 13, 14]. The post-hoc methods target at explaining the black-box models which are fixed or unaccessible. The interpretable methods devote to making transparent prediction from scratch, including not only the predicted label but also the influential subgraph that guides the prediction [14]. In both the post-hoc [10, 11, 37, 39] and the interpretable methods [12, 13, 14], the GNN explainer learns a contribution function $h$ which maps each feature of the input graph into the contribution score to the predicted label. One insight in GNN explanation methods is that, the edge contribution function is more essential to GNN explanation compared with that of the node [11, 13, 14]. For example, when some nodes are selected, it is non-trivial to identify the explanatory subgraph. On the contrary, when the important edges are selected, the correlated endpoints are naturally selected as well. We can naturally identify the explanatory subgraph or further explore the important subset of node features. *In this work, we follow the previous works* [11, 39, 13, 14, 3, 40] *and focus on the contribution of structure features (i.e., edges).* Formally, we learn the contribution function $h$ in terms of each edge in graph $G = (\mathcal{V}, \mathcal{E})$ as follows,

$$\hat{\rho} = h(G), \tag{2}$$

where $\hat{\rho} \in \mathbb{R}^{|\mathcal{E}|}$ and each element in $\hat{\rho}$ is the contribution score of the edge to the task label. Next, a selection module $\mathcal{S}$ as follows is employed to select the edges of the explanatory subgraph $g$, such as the top-$k$ selector [11, 13], the threshold selector [10], and the probabilistic selector [30, 14],

$$g = \mathcal{S}(G, \hat{\rho}). \tag{3}$$

## 3 Methodology

The framework of the proposed $\pi$-GNN is shown in Figure 1, which contains an *explainer pre-training phase* and a *conjoint fine-tuning phase*. In the explainer pre-training phase, we pre-train the $\pi$-GNN explainer over the synthetic dataset PT-Motifs with ground-truth explanations, by taking the binary edge classification as the pretext task. Afterwards, the pre-trained $\pi$-GNN explainer is incorporated with task-specific predictor to identify explanatory subgraphs and provide transparent predictions on different tasks. During the fine-tuning phase, the explainer and the predictor are conjointly optimized. Next, we introduce the $\pi$-GNN model in detail.

### 3.1 Explainer Pre-training Phase

Due to the lack of ground-truth explanation in real-world graph datasets, we first construct a large synthetic dataset with ground-truth explanation called PT-Motifs to support the explainer pre-training.

---

[4]In what follows, we use the term *explainer* and *interpretor* interchangeably [31].

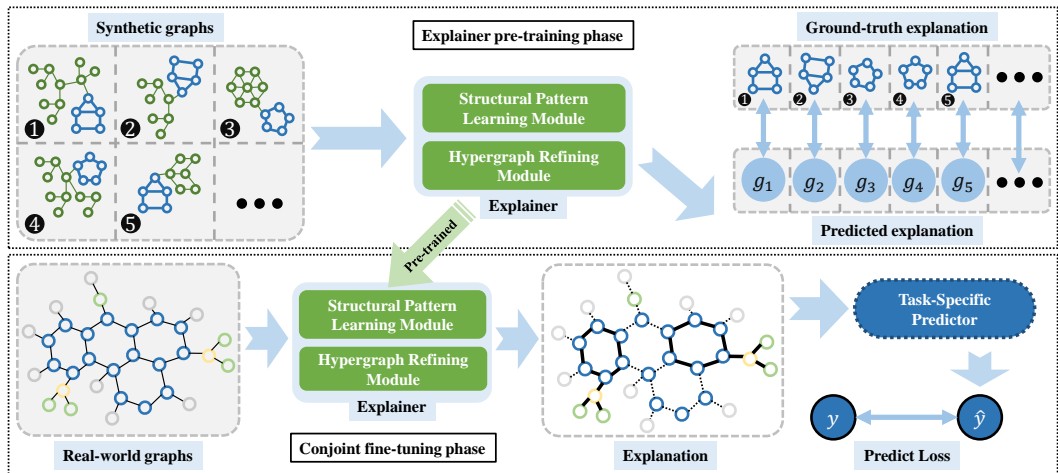

Figure 1: Framework of the $\pi$-GNN model. In the explainer pre-training phase, we use the synthetic graphs with ground-truth explanations to pretrain the $\pi$-GNN explainer. In the fine-tuning phase towards downstream tasks in real-world graphs, the pre-trained explainer and the task-specific predictor are conjointly fine-tuned for transparent and accurate predictions.

Following previous works on generating synthetic graphs [10, 11, 13], each graph $G$ in PT-Motifs dataset consists of one base subgraph $G_b$ and one explanation subgraph $G_e$ (also known as the motif) and the ground-truth task label $y$ which is determined by $G_e$ solely [10, 13]. The shapes of the explanatory subgraphs in PT-Motifs include *Diamond, House, Crane, Cycle, and Star* and the basic shapes are *Clique, Tree, Wheel, Ladder, and the Barabási–Albert Net* [25]. Each combination of base and motif has the same number of samples in PT-Motifs. As shown in Figure 4(e) of Appendix B.2, the degree distribution of PT-Motifs follows the power law function $y = 10^6 \times x^{-1.501}$. See Appendix B.2 for more detailed structural patterns of PT-Motifs. The proposed $\pi$-GNN model distills interpretability from the PT-Motifs dataset during the pre-training phase.

**Structural Pattern Learning Module.** To capture the multiple structural patterns, we propose to parallelize multi-thread of basic pattern-learner $\mathcal{B} = \{B_i | i = 1, 2, \cdots, N\}$. Then, an integrated pattern-learner aggregates them for a more comprehensive and general representation of various graph structural patterns. Specifically, each basic learner $B_i$ identifies a vectorized representation of the structural patterns (such as the degree distribution) and the integrated learner $\Phi$ provides a combination of each individual representation. We formally define the basic pattern-learner as follows.

**Definition 1 (Basic Pattern-Learner).** *Consider a graph $G$ with $n$ nodes, whose adjacency matrix is $\mathbf{A} \in \mathbb{R}^{n \times n}$. The basic pattern-learner $B_i$ projects the adjacency matrix $\mathbf{A}$ into a low-dimensional pattern matrix $\mathbf{Z}_i \in \mathbb{R}^{n \times d}, d < n$, as follows,*

$$\mathbf{Z}_i = B_i(\mathbf{A}). \tag{4}$$

*The basic pattern-learner $B_i$ approaches the low-dimensional pattern matrix $\mathbf{Z}_i$ by maximizing the likelihood of preserving the graph topological structure.*

As a widely adopted technique which provides global views of a graph [41, 42], node embedding can serve as a simple yet effective basic pattern-learner. Inspired by the multi-head attention mechanism for improving the expressive power [43], we parallelize $N$-thread basic pattern-learners to achieve the structural pattern tensor $\mathcal{Z} = [\mathbf{Z}_1, \mathbf{Z}_2, \cdots, \mathbf{Z}_N] \in \mathbb{R}^{N \times v \times d}$ which contains multiple universal structural patterns. For each basic pattern-learner, the adjacency matrix $\mathbf{A}$ is randomly permuted [42, 44]. Afterwards, the integrated learner $\Phi$ defined as follows aggregates the pattern tensor $\mathcal{Z}$ for a more expressive and generalizable pattern representation $\mathbf{Z}_{\text{Int}}$ [45].

**Definition 2 (Integrated Pattern-Learner).** *Given the structural pattern tensor $\mathcal{Z} \in \mathbb{R}^{N \times v \times d}$, the integrated pattern-learner $\Phi$ moves forward to a convex combination $\mathbf{Z}_{\text{Int}} \in \mathbb{R}^{v \times d_{\text{Int}}}$ of each*

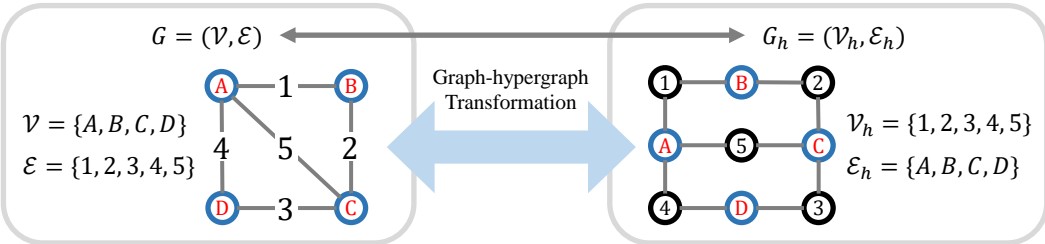

Figure 2: An example of the graph-hypergraph transformation. Considering the 3-degree node $A$ in graph $G$, its corresponding hyper-view is the hyperedge $A$ which connects the hypernodes $\{1, 4, 5\}$.

*individual pattern matrix $\mathbf{Z}_i$ as*

$$\mathbf{Z}_{\text{Int}} = \Phi(\mathcal{Z}) = \sum_{i=1}^{N} \omega_i \mathbf{Z}_i, \tag{5}$$

*where $\Phi$ is an integration function to comprise appropriate structural patterns for diverse graphs.*

Subsequently, $\mathbf{Z}_{\text{Int}}$ is fed into the hypergraph refining module for computing the edge contribution score and exploring explanatory subgraph.

**Hypergraph Refining Module.** With the universal structural patterns extracted from the graphs, we next incorporate it with the edge interactions to identify the explanatory subgraph. Specifically, we first learn the edge structural representation $\mathbf{Z}_E$ from the integrated pattern matrix $\mathbf{Z}_{\text{Int}}$. Taking the edge $e = (i, j)$ as an example, the corresponding structural representation $\mathbf{Z}_E^e$ can be approached from the endpoint representations in $\mathbf{Z}_{\text{Int}}$ as follows,

$$\mathbf{Z}_E^e = f^{(2)}(\mathbf{Z}_{\text{Int}}^i, \mathbf{Z}_{\text{Int}}^j), \tag{6}$$

where $f^{(2)}$ is a 2-variable function and $\mathbf{Z}_{\text{Int}}^i, \mathbf{Z}_{\text{Int}}^j$ are the representations of nodes $v_i, v_j$, respectively.

Since the edge representation $\mathbf{Z}_E$ directly determines the edge contribution score in the explanation, it should express as much information as possible. Theoretically, the expressive power of $\mathbf{Z}_E$ is guaranteed by the following Theorem 1.

**Theorem 1.** *Let $\Sigma_n$ be the set of all adjacency matrix $\mathbf{A}$ with $n$ nodes. Given a graph $G = (\mathcal{V}, \mathcal{E}) \in \Sigma_n, n \geq 2$, let $\Gamma^*(S, \mathbf{A})$ be a most-expressive structural representation of nodes set $S \subseteq \mathcal{V}$ in $G$. $\forall \mathbf{A} \in \Sigma_n$, there exists a most-expressive node representation $\mathbf{Z}^* | \mathbf{A}$ satisfies the relationship as follows,*

$$\Gamma^*(S, \mathbf{A}) = \mathbb{E}_{\mathbf{Z}^*}[f^{(|S|)}((\mathbf{Z}_v^*)_{v \in S}) | \mathbf{A}], \forall S \subseteq \mathcal{V}, \tag{7}$$

*for an appropriate $k$-variable function $f^{(k)}(\cdot)$.*

Theorem 1 defines the upper bound of $\mathbf{Z}_E$ representation ability [44]. The upper bound can be approached according to the Theorem 2 as follows.

**Theorem 2.** *The structural representation of edge $e = (v_i, v_j)$ can be learnt by simply approaching a function $f^{(2)}$ which satisfies $\Gamma(e, \mathbf{A}) = f^{(2)}(\mathbb{E}[(\mathbf{Z}_v)_{v \in \{i,j\}} | \mathbf{A}])$.*

In the hypergraph refining module, we adopt the integrated pattern matrix $\mathbf{Z}_{\text{Agg}}$ as an approximation of the expectation embedding matrix $\mathbb{E}[(\mathbf{Z}_v)_{v \in \{i,j\}} | \mathbf{A}]$ and a 2-layer MLP to fit the transition function $f^{(2)}$ during pre-training phase [44]. See Appendix G for the proofs of Theorems 1 and 2.

Based on the edge structural representation $\mathbf{Z}_E$, a straightforward way is to directly estimate the contribution score of each individual edge [13, 14]. However, such procedure ignores the incorporation of universal patterns with local structural interactions, leading to the missing of the dependencies among the edges. The edges in the explanation are supposed to interact with each other [46], form the coalition, and guide the downstream prediction task better than individuals. We next introduce how to capture the local edge interactions and incorporate them with the universal structural patterns.

Given the graph $G = (\mathcal{V}, \mathcal{E})$, we define its corresponding hypergraph as $G_h = (\mathcal{V}_h, \mathcal{E}_h)$. For a $k$-degree node $v \in \mathcal{V}$, the corresponding hyper-view is a hyperedge connecting $k$ hypernodes.

Similarly, the edge $e \in \mathcal{E}$ becomes a hypernode in the corresponding hypergraph. An example of graph-hypergraph transformation is shown in Figure 2. See Appendix H for the detailed algorithm of graph-hypergraph transformation. During the graph-hypergraph transformation, the edge structural representation is naturally converted to the hypernode structural representation. Therefore, we can capture the local structural interaction by conducting hyperedge message passing. In our implementation, a 2-layer hypergraph convolutional network is employed to capture the edge interaction information and further map the edge structural representations to the edge contribution scores $\hat{\rho}$. Finally, the hypergraph refining is formally represented as,

$$\mathbf{Z}_E^{(1)} = \mathrm{Tanh}(\mathrm{HyperConv}(\mathbf{Z}_E)), \tag{8}$$

$$\hat{\rho} = \sigma(\mathrm{HyperConv}(\mathbf{Z}_E^{(1)})), \tag{9}$$

where the $\mathrm{Tanh}(\cdot)$ function is used for a zero-mean activate value and the $\sigma(\cdot)$ function normalizes the output to a probability value within the range of $[0,1]$. The normalized contribution score in $\hat{\rho}$ measures the probability of each edge belongs to the explanatory subgraph.

Under the supervision of the ground-truth explanation in PT-Motifs, $\pi$-GNN distills the universal interpretability during the pre-training phase. Specifically, for each synthetic graph $G = (\mathcal{V}, \mathcal{E})$, the edges in $\mathcal{E}_P$ that belong to the explanation subgraph are assigned with positive labels and the complementary part $\mathcal{E}_N$ receives a negative label [10, 11]. We denote the ground-truth explanation as $\rho \in \{0,1\}^{|\mathcal{E}|}$. $\pi$-GNN takes $G$ as input and outputs the predicted explanation $\hat{\rho} \in [0,1]^{|\mathcal{E}|}$, which is optimized by the binary cross-entropy loss as follows,

$$L(\hat{\rho}, \rho) = -\sum_{i=1}^{|\mathcal{E}|} [\rho_i \cdot \log\hat{\rho}_i + (1 - \rho_i) \cdot \log(1 - \hat{\rho}_i)]. \tag{10}$$

### 3.2 Conjoint Fine-tuning Phase

During the conjoint fine-tuning phase, we combine the pre-trained $\pi$-GNN explainer with task-specific predictors to identify the explanatory subgraph and make final prediction simultaneously for real-world datasets. Note that the pre-trained $\pi$-GNN explainer is orthogonal to the post-positional predictor. That is, we can implement a predictor with arbitrary architecture as long as it can deal with the graph structure data. Even though the pre-training dataset PT-Motifs belongs to graph classification task, the pre-trained $\pi$-GNN model can be easily generalized to other tasks, such as nodel classification, by simply implementing a node classifier.

Following existing works [11, 13, 14], given the input graph $G$ and the corresponding task label $y$, we introduce a probabilistic sampler $\mathcal{S}$ to comprise the explanatory subgraph $g$ according to the predicted edge probability $\hat{\rho}$ as follows,

$$g = \mathcal{S}(G, \hat{\rho}). \tag{11}$$

Going beyond the probabilistic sampling procedure, the post-positional predictor takes the explanatory subgraph $g$ as input and fits the mapping function to the predicted label $\hat{y}$, by optimizing a task-specific loss function $L_{\mathrm{task}}(\hat{y}, y)$. In addition, we introduce an entropy regularizer [11, 14] in the fine-tuning phase to amplify the gap of each value in $\hat{\rho}$ for sparser explanations as follows,

$$L(\hat{\rho}) = -\sum_{i=1}^{|\mathcal{E}|} [\hat{\rho}_i \cdot \log\hat{\rho}_i + (1 - \hat{\rho}_i) \cdot \log(1 - \hat{\rho}_i)] + ||\hat{\rho}||_1. \tag{12}$$

The overall objective of the fine-tuning phase is to jointly optimize the the task-specific term and the entropy regularizer as follows,

$$L(\hat{y}, \hat{\rho}, y) = L_{\mathrm{task}}(\hat{y}, y) + L(\hat{\rho}). \tag{13}$$

## 4 Experiment

In this section, we conduct extensive experiments to evaluate the performance of $\pi$-GNN by answering the following two questions.

- **RQ1:** How effective is $\pi$-GNN when it is generalized to different graph datasets?
- **RQ2:** How effective is $\pi$-GNN when it is generalized to different graph tasks?

Table 1: Interpretation Performance (ROC-AUC) Comparison. The underlined results highlight the best baselines. The **bold** results mean the $\pi$-GNN or $\pi$-GNN$_{\text{DFT}}$ outperform the best baselines.

| Model | BA-2Motifs | Mutag | MNIST-75sp | Spurious-Motif $b = 0.5$ | $b = 0.7$ | $b = 0.9$ |
|-------|-----------|-------|------------|------|------|------|
| GNNExplainer | $67.35 \pm 3.29$ | $61.98 \pm 5.45$ | $59.01 \pm 2.04$ | $62.62 \pm 1.35$ | $62.25 \pm 3.61$ | $58.86 \pm 1.93$ |
| PGExplainer | $84.59 \pm 9.09$ | $60.91 \pm 17.10$ | $69.34 \pm 4.32$ | $69.54 \pm 5.64$ | $72.33 \pm 9.18$ | $72.34 \pm 2.91$ |
| GraphMask | $92.54 \pm 8.07$ | $62.23 \pm 9.01$ | $73.10 \pm 6.41$ | $72.06 \pm 5.58$ | $73.06 \pm 4.91$ | $66.68 \pm 6.96$ |
| IB-Subgraph | $86.06 \pm 28.37$ | $91.04 \pm 6.59$ | $51.20 \pm 5.12$ | $57.29 \pm 14.35$ | $62.89 \pm 15.59$ | $47.29 \pm 13.39$ |
| DIR | $82.78 \pm 10.97$ | $64.44 \pm 28.81$ | $32.35 \pm 9.39$ | $78.15 \pm 1.32$ | $77.68 \pm 1.22$ | $49.08 \pm 3.66$ |
| GIN-GSAT | $98.74$ $\pm 0.55$ | $99.60$ $\pm 0.51$ | $83.36 \pm 1.02$ | $78.45 \pm 3.12$ | $74.07 \pm 5.28$ | $71.97 \pm 4.41$ |
| PNA-GSAT | $93.77 \pm 3.90$ | $99.07 \pm 0.50$ | $84.68$ $\pm 1.06$ | $83.34$ $\pm 2.17$ | $86.94$ $\pm 4.05$ | $88.66$ $\pm 2.44$ |
| $\pi$-GNN | $\mathbf{99.33} \pm 0.63$ | $\mathbf{99.81} \pm 0.17$ | $\mathbf{92.77} \pm 0.80$ | $\mathbf{93.24} \pm 0.72$ | $\mathbf{96.92} \pm 0.85$ | $\mathbf{96.39} \pm 0.92$ |
| $\pi$-GNN$_{\text{DFT}}$ | $93.19 \pm 1.48$ | $95.29 \pm 0.67$ | $\mathbf{85.18} \pm 1.08$ | $\mathbf{86.29} \pm 2.22$ | $\mathbf{87.43} \pm 2.47$ | $\mathbf{89.64} \pm 2.26$ |

## 4.1 Experimental Settings

In the experiment, we use two popular synthetic datasets [10, 11, 13, 14] and four real-world datasets of graph classification tasks. The details of dataset characteristics and statistics are summarized in Appendix B. A brief introduction to the datasets is as follows.

- **Synthetic Datasets.** BA-2Motifs [10] and Spurious-Motif [13] are two widely-used synthetic datasets to evaluate the interpretation performance of the GNN explanation methods.

- **Real-world Datasets.** We use four real-world datasets, the superpixel graph dataset MNIST-75sp [40], the sentiment analysis dataset Graph-SST2 [16], and two chemical molecule datasets Mutag [18] and Ogbg-Molhiv [47]. Note that, in the MNIST-75sp dataset, the subgraph with nonzero pixel values is regarded as the ground-truth explanation [14]; in the Mutag dataset, -NO$_2$ and -NH$_2$ functional groups in mutagen graphs are labelled as the ground-truth explanation [28]. Hence, we use the MNIST-75sp and Mutag datasets for both the interpretation and the prediction evaluations.

We extensively compare $\pi$-GNN with the following two types of baselines:

- **Interpretation Baselines.** We compare the interpretation performance with both the post-hoc explanation methods including GNNExplainer [10], PGExplainer [11], and GraphMask [39] and the intrinsic interpretable methods including DIR [13], IB-subgraph [12], GIN-GSAT, and PNA-GSAT [14]. Following the standard setting, the evaluation metric is the explanation ROC-AUC [13, 14].

- **Prediction Baselines.** We compare the perdiction performance with the powerful GNN models including GIN [4] and PNA [32] and the intrinsic interpretable methods including DIR, IB-subgraph, GIN-GSAT, and PNA-GSAT. For the OGBG-Molhiv dataset, we use the classification ROC-AUC as the prediction metric [47]. For all the other dataset, we report the classification accuracy [13].

Additionally, we report the performance of $\pi$-GNN which **directly fine-tunes** on downstream datasets without pre-training (denoted as the $\pi$-GNN$_{\text{DFT}}$), to investigate the effectiveness of pre-training phase. All the results are averaged over 10-times evaluation with different random seeds. Architecture of the downstream GNN predictors used in $\pi$-GNN and $\pi$-GNN$_{\text{DFT}}$ are reported in Appendix C.

## 4.2 Main Results (RQ1)

To investigate the effectiveness of $\pi$-GNN, we compare the interpretation and the prediction performance with the SOTA interpretation and prediction baselines. See Appendix C for the pre-training and fine-tuning details. The overall interpretation performance and prediction performance are summarized in Table 1 and Table 2, respectively. We conclude the following observations:

- **$\pi$-GNN significantly outperforms the leading GNN explanation methods.** Specifically, for the Spurious-Motif dataset, $\pi$-GNN surpasses DIR by 27.21% on average and by 47.31% at most. Compared with the best baselines, i.e., the GSAT with a 4-layer PNA encoder, $\pi$-GNN improves the interpretation performance by 9.20% on average. However, we merely employ the combination of a truncated SVD embedding and a 2-layer MLP as the encoder, which convincingly demonstrates the effectiveness of our proposed pre-training phase. Moreover, as the degree of spurious correlation in

Table 2: Prediction Performance (Acc) Comparison. The underlined results highlight the best baselines. The **bold** results mean the $\pi$-GNN or $\pi$-GNN$_{\text{DFT}}$ outperform the best baselines.

| Model | Molhiv(AUC) | Graph-SST2 | MNIST-75sp | Spurious-Motif | | |
| | | | | $b = 0.5$ | $b = 0.7$ | $b = 0.9$ |
| --- | --- | --- | --- | --- | --- | --- |
| GIN | $76.69 \pm 1.25$ | $82.73 \pm 0.77$ | $95.74 \pm 0.36$ | $39.87 \pm 1.30$ | $39.04 \pm 1.62$ | $38.57 \pm 2.31$ |
| PNA | $78.91 \pm 1.04$ | $79.87 \pm 1.02$ | $87.20 \pm 5.61$ | $68.15 \pm 2.39$ | $\underline{66.35} \pm 3.34$ | $\underline{61.40} \pm 3.56$ |
| IB-Subgraph | $76.43 \pm 2.65$ | $\underline{82.99} \pm 0.67$ | $93.10 \pm 1.32$ | $54.36 \pm 7.09$ | $48.51 \pm 5.76$ | $46.19 \pm 5.63$ |
| DIR | $76.34 \pm 1.01$ | $82.32 \pm 0.85$ | $88.51 \pm 2.57$ | $45.49 \pm 3.81$ | $41.13 \pm 2.62$ | $37.61 \pm 2.02$ |
| GIN-GSAT | $76.47 \pm 1.53$ | $82.95 \pm 0.58$ | $\underline{96.24} \pm 0.17$ | $52.74 \pm 4.08$ | $49.12 \pm 3.29$ | $44.22 \pm 5.57$ |
| PNA-GSAT | $\underline{80.24} \pm 0.73$ | $80.92 \pm 0.66$ | $93.96 \pm 0.92$ | $\underline{68.74} \pm 2.24$ | $64.38 \pm 3.20$ | $57.01 \pm 2.95$ |
| $\pi$-GNN | $\mathbf{80.86} \pm 0.61$ | $\mathbf{88.05} \pm 0.43$ | $\mathbf{96.89} \pm 0.20$ | $\mathbf{74.67} \pm 0.63$ | $\mathbf{77.52} \pm 0.77$ | $\mathbf{77.46} \pm 0.96$ |
| $\pi$-GNN$_{\text{DFT}}$ | $79.71 \pm 1.08$ | $\mathbf{83.48} \pm 1.20$ | $92.89 \pm 0.95$ | $\mathbf{70.78} \pm 1.63$ | $\mathbf{71.02} \pm 1.43$ | $\mathbf{72.61} \pm 1.75$ |

Spurious-Motif (i.e., the parameter $b$) increasing, the performance of $\pi$-GNN does not decrease as most of the baselines. The post-hoc explainers, including GNNExplainer, PGExplainer, and GraphMask seems to be robust in terms of the spurious correlation, but their interpretability is limited by the fixed prediction model. Although DIR and GSAT introduce complex mechanism for mitigating the spurious correlation, $\pi$-GNN still performs better. We ascribe this superiority to the generalizable knowledge which is distilled from the pre-training phase over large dataset with ground-truth explanations. For the BA-2Motifs and the Mutag datasets, $\pi$-GNN using a more simpler encoder architecture also achieves comparable performance. For the MNIST-75sp dataset, $\pi$-GNN surpasses the best baseline PNA-GSAT by 4.50%. Such top-tier performance strongly validates the effectiveness of $\pi$-GNN explainer.

- **$\pi$-GNN also achieves better prediction performance than the baselines.** Overall, $\pi$-GNN outperforms all the prediction baselines consistently by a significant margin. Specifically, for the Spurious-Motif dataset, $\pi$-GNN outperforms the best baselines by 11.05% on average, which is a significant improvement. If we restrict the baselines into interpretable GNNs, the performance boost is much more significant (by 13.17% on average and up to 20.45%). For the Graph-SST2 dataset, $\pi$-GNN outperforms the black-box predictor (i.e., GIN and PNA), as well as the SOTA interpretable methods IB-Subgraph, which optimizes the explanatory subgraph based on the information bottleneck principle. Note that, we only implement a 2-layer GCN with global mean pooling function as the Graph-SST2 predictor. We credit such outperformance to the pre-training phase, from which the universal patterns of the graphs are potentially distilled. For the MNIST-75sp dataset, $\pi$-GNN achieves comparable prediction accuracy with the GSAT methods, but our interpretation ROC-AUC exceeds GSAT by 8.09%. Additionally, one can notice that the black-box predictors are insensitive to the spurious correlation while the interpretable baselines deteriorate obviously. This phenomena may indicate that insufficient interpretability conflicts with the chase of prediction accuracy, but a powerful interpretor is quite helpful to the subsequent predictor.

- **The pre-training phase advances the interpretor in terms of both interpretation and perdiction performance.** As shown in Table 1 and Table 2, the $\pi$-GNN with pre-training phase significantly improves both the interpretation and the prediction performance, compared with the reduced variant $\pi$-GNN$_{\text{DFT}}$. For the interpretation ROC-AUC, $\pi$-GNN outperforms the reduced variant by 6.91% on average. This suggests that the universal structural patterns distilled in the pre-training phase can indeed generalize to various downstream tasks and improve the interpretability. Compared with the reduced variant, $\pi$-GNN consistently provides much stabler interpretation with smaller variance. Additionally, one can observe that the reduced variant $\pi$-GNN$_{\text{DFT}}$ still surpasses the best baselines on some real-world datasets, such as the interpretation performance on MNIST-75sp dataset and the prediction performance on Graph-SST2 dataset. This may imply that the edge interaction captured by $\pi$-GNN$_{\text{DFT}}$ is able to identify the influential subgraphs more accurately.

Moreover, we present the explanatory visualization, the investigate of different pre-training datasets and the hyper-parameter analysis in Appendix D, E and F, respectively.

### 4.3 Inter-Task Generalization Performance (RQ2)

To further investigate whether the universal structural patterns behind different tasks is common, we next study the generalization ability across different tasks. Specifically, we evaluate the $\pi$-GNN model that is pre-trained over graph classification dataset PT-Motifs on the explanation task of node

Table 3: Inter-Task Interpretation Performance (ROC-AUC) Camparison. The **bold** font highlights the best results and the underlined results highlight the second best method.

| Model | BA-Shapes | BA-Community | Tree-Cycles | Tree-Grid |
|---|---|---|---|---|
| GRAD | 88.20 | 75.00 | 90.50 | 61.20 |
| Attention | 81.50 | 73.90 | 82.40 | 66.70 |
| GNNExplainer | 92.50 | 83.60 | 94.80 | 87.50 |
| PGExplainer | **96.30** $\pm$ 1.10 | 94.50 $\pm$ 1.90 | **98.70** $\pm$ 0.70 | **90.70** $\pm$ 1.40 |
| $\pi$-GNN | 94.78 $\pm$ 0.32 | **94.67** $\pm$ 1.50 | 95.19 $\pm$ 0.88 | 90.11 $\pm$ 1.10 |
| $\pi$-GNN$_{\text{DFT}}$ | 93.17 $\pm$ 0.40 | 92.15 $\pm$ 1.61 | 92.53 $\pm$ 1.81 | 88.62 $\pm$ 1.87 |

classification datasets. The pre-trained $\pi$-GNN model and the reduced variant are both equipped with a node classifier. Following existing works, we use four widely-used synthetic node classification datasets [10], namely BA-Shapes, BA-Community, Tree-Cycles and Tree-Grid, whose detailed statistics are presented in Appendix B. The main result of inter-task explanation in Table 3 shows that $\pi$-GNN achieves comparable performan with SOTA node classification explainers. This evidence is accordant with our basic premise that the universal structural patterns can generalize across datasets of different tasks. For the BA-Community dataset, $\pi$-GNN even supasses PGExplainer with smaller varience. This may be because the community structure in the BA-Community graph conforms more to the universal patterns embedded in $\pi$-GNN.

### 4.4 Ablation Study

As shown in Figure 3, we conduct ablation study on $\pi$-GNN, by evaluate the performance of its three variants. First, **$\pi$-GNN-SPL** substitutes the structural pattern learning module with $N$-thread GNN encoders. Second, **$\pi$-GNN-HPR** removes the hypergraph refining module and directly calculates the edge contribution score. Third, we simultaneously conduct the two ablations above and denote it as **$\pi$-GNN-ALL**. Additionally, we report the performance of the variant without pre-training phase.

Specifically, when removing the structural pattern learning module, the interpretation performance on Mutag decreases by 3.85% on average and the prediction performance on Graph-SST2 decreases by 1.08% on average. For the variant $\pi$-GNN-HPR, the interpretation and prediction performance decreases by 6.56% and 1.84%, respectively. When we remove the two modules simultaneously, the interpretation and prediction performance decreases by 7.56% and 2.16%, respectively. The ablation study on the two modules demonstrates their effectiveness in capturing universal structural patterns and identifying the explanatory subgraphs. Moreover, for all the variants, $\pi$-GNN consistently outperforms the variant without pre-training phase by 5.59% in terms of ROC-AUC on Mutag and 2.84% in terms of accuracy on Graph-SST2.

## 5 Related Work

**Intrinsic interpretable GNNs.** The leading interpretable GNNs [12, 13, 14] usually consist of an explainer module and a predictor module. The prepositional explainer takes the raw graph as input and outputs the explanation subgraph. The subsequent predictor calculates the prediction strictly relying on the explanation. Graph neural networks with attention mechanism are regarded as the initial interpretable GNNs [13, 14], such as graph attention network [3], self-attention graph pooling [40], where the learned weights can be interpreted as the importance of certain features. Recently, invariant learning is introduced to construct intrinsic interpretable GNNs [48, 49], such as DIR [13]. It argues that augmenting training data with causal intervention may assist explainer to distinguish the causal and non-causal parts. Besides, interpretable GNNs based on the information bottleneck principle [50], such as IB-Subgraph [12] and GSAT [14], are proposed to constraint the information flow from the input graph to the prediction, where the label-relevant graph components will be kept while the label-irrelevant ones are reduced.

**Pre-training on Graphs.** The research focus of current graph pre-training is the graph representation learning problem, whose objective is to learn a generic encoder $f(\mathbf{A}, \mathbf{X})$ for various downstream tasks [51, 52]. The development of graph pre-training can be broadly divided into pre-trained graph embeddings [41, 53, 42] and pre-trained graph encoders [23, 22, 24, 21]. Pre-trained graph embedding models aim to provide good graph embeddings for various tasks, while the models themselves are no

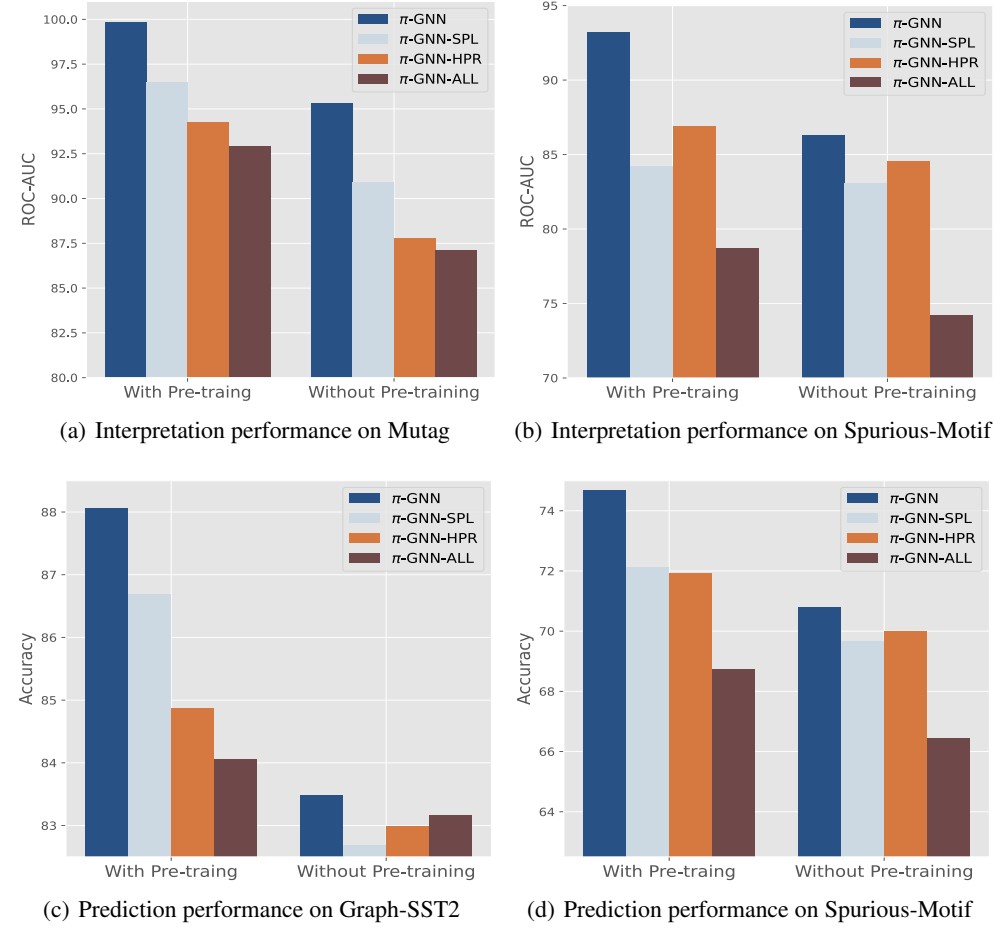

(a) Interpretation performance on Mutag

(b) Interpretation performance on Spurious-Motif

(c) Prediction performance on Graph-SST2

(d) Prediction performance on Spurious-Motif

Figure 3: The ablation study of $\pi$-GNN on Mutag, Graph-SST2, and Spurious-Motif datasets.

longer needed to the tasks. DeepWalk [41] explores the graph embeddings by conducting random walks over graphs to generate node sequences which contain the co-occurrence relationship. Further, Node2vec [42] defines a flexible node neighborhood and proposes a biasd random walk process. The goal of pre-trained graph encoder models is a generic encoder model which can deal with different tasks. Gpt-GNN [23] pre-trains a 5-layer GIN encoder across the graph-level and node-level tasks. GCC [22] introduces the contrastive learning framework to pre-train the encoder over subgraph discrimination task. However, the precursor graph pre-training works are not designed to the graph explanation problem and can not be directly applied to the pre-training interpretable GNNs.

## 6 Conclusion

In this work, we for the first time investigated the universal interpretation problem in graph data and proposed the Pre-training Interpretable Graph Neural Network named $\pi$-GNN. $\pi$-GNN is able to work well on different types of graphs and downstream tasks. $\pi$-GNN was pre-trained over a constructed large synthetic graph dataset with ground-truth explanations to distill the generalizable interpretability. Then, $\pi$-GNN was fine-tuned on downstream tasks. Technically, we proposed an intergrated embedding module to capture and integrate multiple graph structural patterns for more generalizable representations. A hypergarph refining module was aslo proposed to incorporate the universal patterns with local interaction for more faithful explanatory subgraphs identification. Extensive experiments on different datasets and tasks demonstrated the promising generalizable interpretability as well as prediction performance of $\pi$-GNN.

## Acknowledgement

This research was funded by the National Science Foundation of China (No. 62172443), Open Project of Xiangjiang Laboratory (22XJ03010, 22XJ03005), the Science and Technology Major Project of Changsha (No. kh2202004), Hunan Provincial Natural Science Foundation of China (No. 2022JJ30053), and the High Performance Computing Center of Central South University.

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

# A Notations

In Table 4, we give the mathematical symbols and their descriptions used in this paper for clarity.

Table 4: Symbols and their descriptions.

| Symbol | Description |
|---|---|
| $G, \mathcal{V}, \mathcal{E}$ | Graph instance, node set, edge set |
| $\mathbf{X}$ | Node feature matrix |
| $\mathbf{X}_E$ | Edge feature matrix |
| $\mathbf{A}, A_{ij}$ | Adjacency matrix, element at the $i$-th row, $j$-th column of $\mathbf{A}$ |
| $v, v_i; e, e_i$ | Node instance; edge instance |
| $\mathcal{N}(v)$ | Neighborhood of node $v$ |
| $\mathbf{W}$ | Trainable parameter matrix |
| $F(\cdot), \sigma(\cdot)$ | Non-linear activation function, Sigmoid function |
| $h$ | Edge contribution function |
| $\rho, \hat{\rho}$ | Ground-truth explanation, predicted explanation |
| $\mathcal{S}$ | Selection module |
| $g$ | Explanatory subgraph |
| $\mathcal{B}, B_i$ | Set of basic pattern-learners, basic pattern-learner |
| $N$ | Number of basic pattern-learners |
| $\mathbf{Z}_i, \mathcal{Z}$ | Pattern matrix, pattern tensor |
| $\Phi$ | Integrated pattern-learner |
| $\mathbf{Z}_{\text{Int}}$ | Integrated pattern representation |
| $\mathbf{Z}_E$ | Edge structural representation |
| $S$ | Subset of node |
| $f^{(2)}$ | Node to edge transition function |
| $G_h, \mathcal{V}_h, \mathcal{E}_h$ | Hypergraph instance, hypernode set, hyperedge set |
| $\mathcal{E}_P, \mathcal{E}_N$ | Positive (Negative) edge set |
| $b$ | Degree of spurious correlation in Spurious-Motif dataset |

# B Datasets

We introduce the datasets used in the main experiment as follows.

- **BA-2Motifs** [10] is a synthetic dataset with binary graph classes for evaluating the interpretation performance. House motifs and cycle motifs decide the graph labels and serve as the ground-truth explanations of the two classes, respectively.

- **Spurious-Motif** [13] is a synthetic dataset with three graph classes for evaluating both the interpretation and the prediction performance. Each class corresponds to a particular motif which is regarded as the ground-truth explanation. During the synthesis process, spurious correlations, including the unbalanced base sampling and the scale drift, are injected into the training dataset. A hyper-parameter $b$ controls the degree of spurious correlation, usually set as $\{0.5, 0.7, 0.9\}$.

- **Mutag** [18] is a chemical molecule dataset with binary graph classes which represents mutagenic property. Benefiting from domain knowledge, -NO$_2$ and -NH$_2$ functional groups in mutagen graphs are labelled as the ground-truth explanation. Therefore, we employ the Mutag dataset for the evaluation of interpretation and prediction performance.

- **MNIST-75sp** [40] converts the MNIST image dataset into a superpixel graph dataset with ten graph classes. The nodes are superpixels, while edges are the spatial distance between the related endpoints. The subgraph with nonzero pixel values is regarded as the ground-truth explanation. Hence, the Mnist-75sp is used for both the interpretation and the prediction evaluations.

- **Graph-SST2** [16] is a sentiment analysis dataset with binary graph labels, where each sequence in SST2 dataset is transformed to a graph. Since no ground-truth explanations are available, we evaluate the prediction performance only.

- **OGBG-Molhiv** [47] is a molecular dataset with binary graph labels according to the inhibition effect on HIV virus replication, where nodes are atoms, and edges are chemical bonds. As there are no ground-truth explanations, we merely evaluate the prediction performance.

## B.1 Statistical Characteristics

We show the detailed statistics of the datasets in Table 5.

Table 5: Detailed statistics of graph classification datasets in main experiment.

|  | BA-2Motifs | | | Spurious-Motif | | | Mutag | | |
|---|---|---|---|---|---|---|---|---|---|
|  | Train | Val | Test | Train | Val | Test | Train | Val | Test |
| #Classes | | 2 | | | 3 | | | 2 | |
| #Graphs | 400 | 400 | 200 | 9,000 | 3,000 | 6,000 | 1000 | 1000 | 951 |
| Avg. #Nodes | 25.0 | 25.0 | 25.0 | 25.4 | 26.1 | 88.7 | 30.1 | 30.1 | 30.2 |
| Avg. #Edges | 50.9 | 51.0 | 50.9 | 35.4 | 36.2 | 131.1 | 61.3 | 60.2 | 61.2 |

|  | MNIST-75sp | | | Graph-SST2 | | | OGBG-Molhiv | | |
|---|---|---|---|---|---|---|---|---|---|
|  | Train | Val | Test | Train | Val | Test | Train | Val | Test |
| #Classes | | 10 | | | 2 | | | 2 | |
| #Graphs | 20,000 | 5,000 | 10,000 | 28,237 | 3,147 | 12,305 | 32,901 | 4,113 | 4,113 |
| Avg. #Nodes | 66.8 | 67.3 | 67.0 | 17.7 | 17.3 | 3.45 | 25.3 | 27.8 | 25.3 |
| Avg. #Edges | 539.3 | 545.9 | 540.9 | 33.3 | 33.5 | 4.89 | 54.1 | 61.1 | 55.6 |

We show the detailed statistics of datasets that used in the inter-task evaluation in Table 6.

Table 6: Detailed statistics of node classification datasets in inter-task experiment.

|  | BA-Shapes | BA-Community | Tree-Cycles | Tree-Grid |
|---|---|---|---|---|
| #Classes | 4 | 8 | 2 | 2 |
| #Nodes | 700 | 1,400 | 871 | 1,231 |
| #Edges | 4,110 | 8,920 | 1,950 | 3,410 |

## B.2 Structural Patterns

**Degree Distribution.** As shown in Figure 4, we visualize the degree distribution of the four real-world datasets, i.e., Mutag, MNNIST-75sp, Ogbg-Molhiv, and Graph-SST2, and our synthetic scale-free dataset PT-Motifs. The power-law like degree function $D(x) = cx^{-\alpha}$ of each real-world dataset is formally represented as follows, optimized by the ordinary least square error. Here $D(x)$ represents the number of $x$-degree nodes and $R^2$ is the coefficient of determination.

- **Mutag:** $D(x) = 54447x^{-1.537}, R^2 = 0.9015$.
- **MNIST-75sp:** $D(x) = 10^7 x^{-5.913}, R^2 = 0.6713$.
- **Ogbg-Molhiv:** $D(x) = 3 \times 10^6 x^{-6.039}, R^2 = 0.6828$.
- **Graph-SST2:** $D(x) = 8 \times 10^6 x^{-4.988}, R^2 = 0.8491$.
- **PT-Motifs:** $D(x) = 10^6 x^{-1.501}, R^2 = 0.9677$.

In Table 7, we report the average value of the four structural patterns over each individual graphs in the datasets, including

- **Transitivity** is defined as the fraction of all possible triangles present in the given graph $G$. It measures the tendency of connections or relationships between nodes to form triangles or triplets and captures the clustering characteristics in a given graph. Let *triads* be the structure that two edges with a shared node, then the transitivity can be formally represented as follows,

$$\text{Transitivity} = 3\frac{\#\text{Triangles}}{\#\text{Triads}}, \tag{14}$$

where $\#\text{Triangles}$ and $\#\text{Triads}$ are the number of triangles and triads in $G$, respectively.

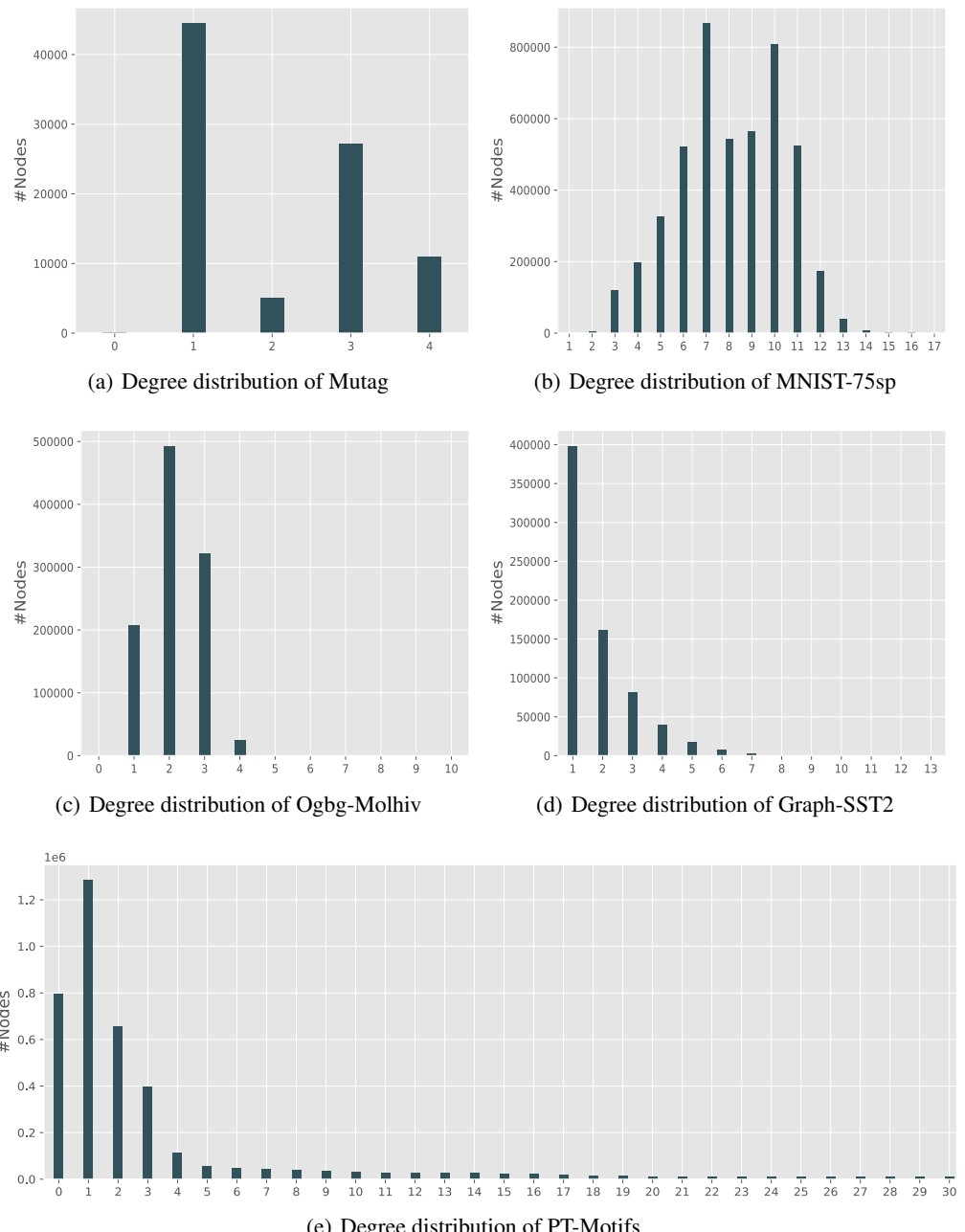

Figure 4: The degree distribution of four real-world graph datasets (Mutag, MNIST-75sp, Ogbg-Molhiv, and Graph-SST2) and our synthetic dataset PT-Motifs. The $x$ axis is the node degree.

- **Assortativity** is defined as the similarity of connections in the given graph $G$ with respect to the node degree. It measures the preference of nodes to connect with nodes of similar or different characteristics. Assortativity helps uncover the underlying patterns of connectivity and provides insights into the organization and functioning of complex networks. The average assortativity of the given graph $G$ is the mean value over all nodes.

- **Efficiency** of a pair of nodes is defined as the multiplicative inverse of the shortest path distance between the endpoints in the given graph $G$. The average global efficiency of a graph is the average efficiency of all pairs of nodes.

- **Clustering** of the node $v$ is defined as the fraction of possible triangles through that node. The corresponding clustering coefficient $c_u$ can be formulated as,

$$c_v = 2\frac{\#\text{Triangles}_u}{\deg_u(\deg_u - 1)}, \tag{15}$$

where $\#\text{Triangles}_u$ is the number of triangles through node $u$ and $\deg_u$ is degree of node $u$. The average clustering coefficient $C$ of the given graph $G$ with $n$ nodes is defined as follows,

$$C = \frac{1}{n}\sum_{v \in G} c_v. \tag{16}$$

Table 7: Structural patterns of the PT-Motifs dataset and four real-world datasets.

|  | PT-Motifs | Mutag | MNIST-75sp | Molhiv | Graph-SST2 |
|---|---|---|---|---|---|
| Transitivity | 0.3620 | 0.0013 | 0.5005 | 0.0021 | 0 |
| Avg Assortativity | 0.1525 | -0.4196 | 0.3235 | -0.2678 | -0.6269 |
| Avg Efficiency | 0.5044 | 0.3261 | 0.3977 | 0.3243 | 0.5491 |
| Avg Clustering | 0.4185 | 0.0010 | 0.5408 | 0.0020 | 0 |

For an intuitive understanding of the graph structural patterns, we visualize the distribution of *node pair efficiency* and *node assortativity* in Figure 5 and Figure 6, respectively. One can notice that the efficiency distributions of PT-Motifs, Mutag, MNIST-75sp, and Molhiv all follow a bell-shaped curve, while that of Graph-SST2 is a little different. Similarly, we found that the node assortativity distributions of Mutag, MNIST-75sp, Molhiv, and PT-Motifs all have one main peak which largely surpasses the adjacent values, while the node assortativity of Graph-SST2 dataset has four peaks.

Overall, the structural patterns above, including the degree distribution, the transitivity, the assortativity, the efficiency, and the clustering, are universal and generalizable across different datasets. However, the expression degree of different structural patterns may differ largely in different datasets. As shown above, the Graph-SST2 dataset strictly follows the power law shaped degree distribution, but its node pair efficiency and node assortativity distributions are different from the other three real-world datasets. Therefore, we propose the structural pattern learning module in $\pi$-GNN, to capture multiple structural patterns and integrate them for a more universal and generalizable representation.

## C Experimental Details

We first present the details of the pre-training phase over the synthetic PT-Motifs dataset. The PT-Motifs dataset is split into the training set of 50,000 graphs, the validation set of 10,000 graphs, and the testing set of 20,000 graphs. Each graph class has equal number of instances in the three sets. During the pre-training phase, the batchsize is set as $\{32, 64, 128, 256\}$ and the learning rate is set as $\{10^{-3}, 5 \times 10^{-3}, 10^{-4}, 10^{-5}, 10^{-6}\}$. The pre-training epoch is set as $\{20, 40, 60, 80\}$. We select the pre-trained $\pi$-GNN model according to the validation performance. During the pre-training and the fine-tuning phases, we use the Adam optimizer.

As shown in Tables 8 and 9, we present both the downstream predictor architecture and the fine-tuning details of the graph classification datasets and the node classification datasets, respectively. For all the graph classification datasets, we use the global mean pooling as the pooling function. Specially, in the Molhiv predictor, we introduce the virtual node and weighted loss tricks to mitigate the class-imbalance issue (39,684 negative examples and only 1,443 positive examples). All experiments are conducted on a single NVIDIA GeForce 3090 GPU (24GB).

## D Explanatory Visualization

For an intuitive understanding on the $\pi$-GNN explanation, we present a few visualized explanation results of the Graph-SST2 dataset in Figure 7. Overall, $\pi$-GNN has the ability to highlight the influential phrases that directly express the positive or negative sentiments in the sentence. Specifically, $\pi$-GNN correctly allocates large weights to the positive words *"a legend"* in Figure 7(a), as well as *"bewilderingly brilliant"* and *"entertaining"* in Figure 7(c). Furthermore, the negative word, such

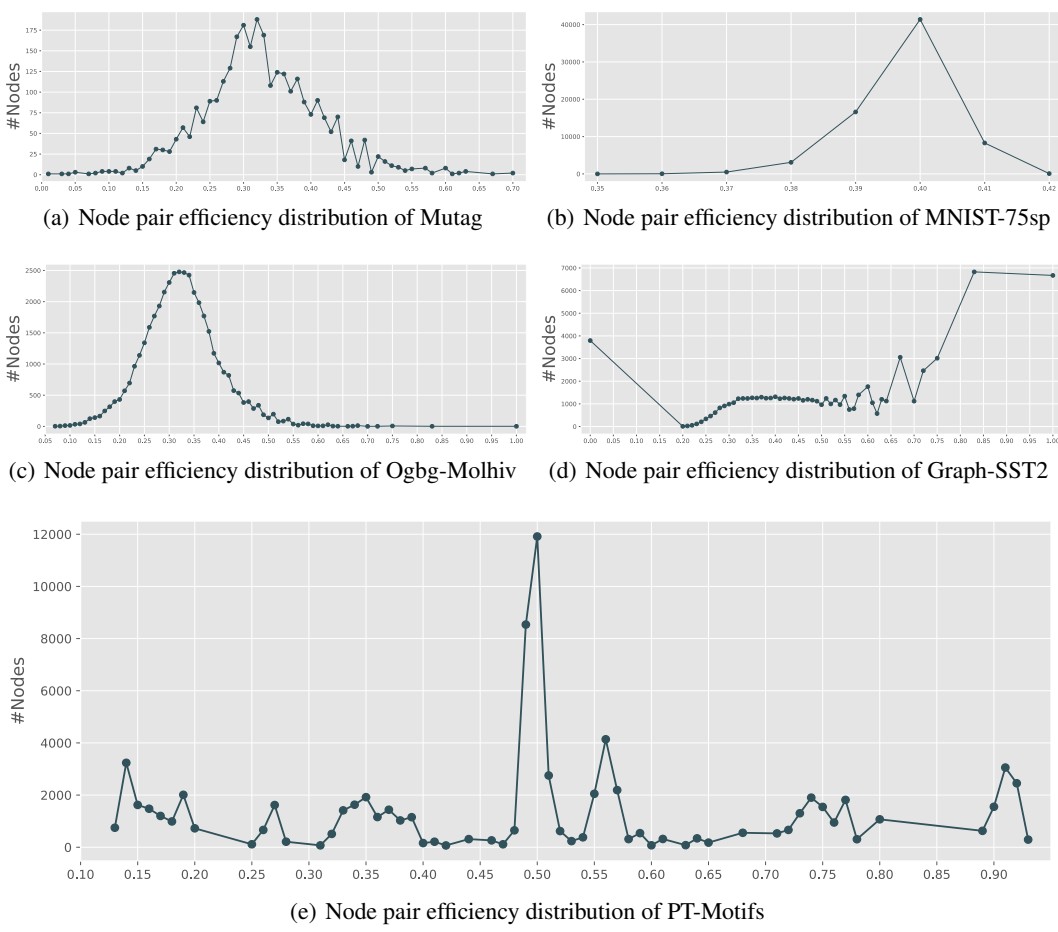

Figure 5: The node pair efficiency distribution of four real-world graph datasets (Mutag, MNIST-75sp, Ogbg-Molhiv, and Graph-SST2) and the synthetic dataset PT-Motifs.

Table 8: Predictor architecture and fine-tuning details of graph classification datasets.

|  | BA-2Motifs | Spurious-Motif | Mutag | MNIST-75sp | Molhiv | Graph-SST2 |
|---|---|---|---|---|---|---|
| Backbone | GCN | GCN | GCN | GIN | GIN | GCN |
| Layers | 1 | 1 | 1 | 2 | 2 | 1 |
| Batchsize | 256 | 128 | 64 | 256 | 128 | 32 |
| Learning rate | $8 \times 10^{-4}$ | $4 \times 10^{-4}$ | $1 \times 10^{-4}$ | $1 \times 10^{-3}$ | $1 \times 10^{-3}$ | $1 \times 10^{-4}$ |
| Epochs | 30 | 30 | 30 | 50 | 40 | 40 |

as *"absurd lengths"* and *"skip dreck"* in Figure 7(b) and Figure 7(d), are identified by $\pi$-GNN for a transparent sentiment prediction. The visualized explanation again demonstrates the effectiveness of $\pi$-GNN: (1) the pre-trained $\pi$-GNN interpretor can faithfully extract the most vital subgraph that contains the label-relevant information; and (2) the pre-trained $\pi$-GNN interpretor is able to cooperate with specific downstream predictor for both accuracy and interpretability. See Figure 7 for more visualized results of the explanations.

# E Supplement Experiment

To further investigate the impact of the pre-training dataset on the final results, we conduct emprical studies on the size and the imbalance degree of the pre-training dataset.

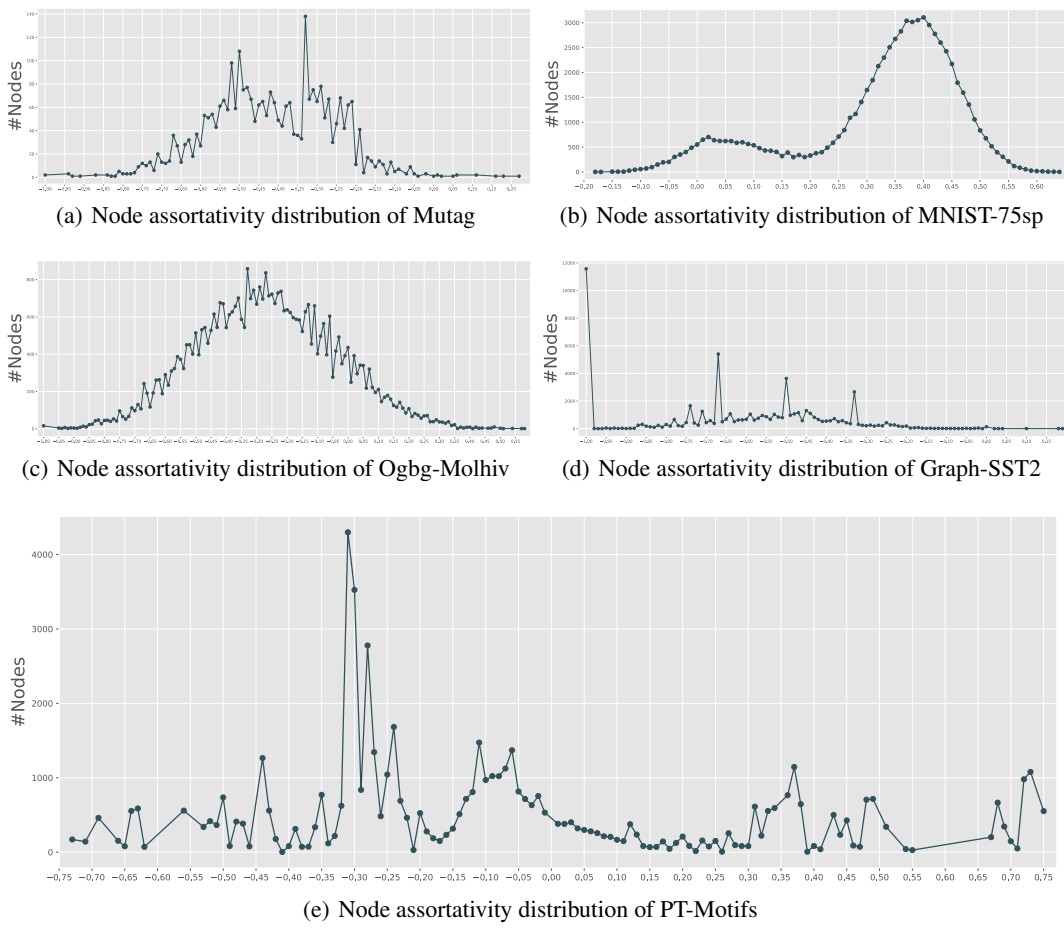

Figure 6: The node assortativity distribution of four real-world graph datasets (Mutag, MNIST-75sp, Ogbg-Molhiv, and Graph-SST2) and our synthetic dataset PT-Motifs.

Table 9: Predictor architecture and fine-tuning details of node classification datasets.

|  | BA-Shapes | BA-Community | Tree-Cycles | Tree-Grid |
|---|---|---|---|---|
| Backbone | GCN | GCN | GCN | GCN |
| Layers | 3 | 3 | 3 | 3 |
| Learning rate | $1 \times 10^{-1}$ | $1 \times 10^{-1}$ | $1 \times 10^{-2}$ | $5 \times 10^{-2}$ |
| Epochs | 20 | 20 | 20 | 20 |

First, compared with PT-Motfis (80,000 graphs) in the main experiment, we generate another two datasets with different sizes. In detail, PT-Motifs-M with 50,000 graphs and PT-Motifs-S with 10,000 graphs represent the middle-level and the small-level pre-training datasets, respectively. The split ratio of the training, validation, and testing set is $\{0.7, 0.1, 0.2\}$ in PT-Motifs-M and PT-Motifs-S. As shown in Table 10, the results show that even the PT-Motifs-S is able to outperform the model without pre-training. Moreover, we can notice that a large pre-training dataset can indeed improve the performance more significantly than that with small size.

Furthermore, it is possible that if the synthetic pre-training dataset is imbalanced, the performance improvement on downstream tasks will degrade, in terms of both the interpretation and prediction. To further investigate this issue, we have added supplement experiments on imbalanced pre-training dataset. Following existing works [13, 11], to generate the imbalanced datasets, we sample the

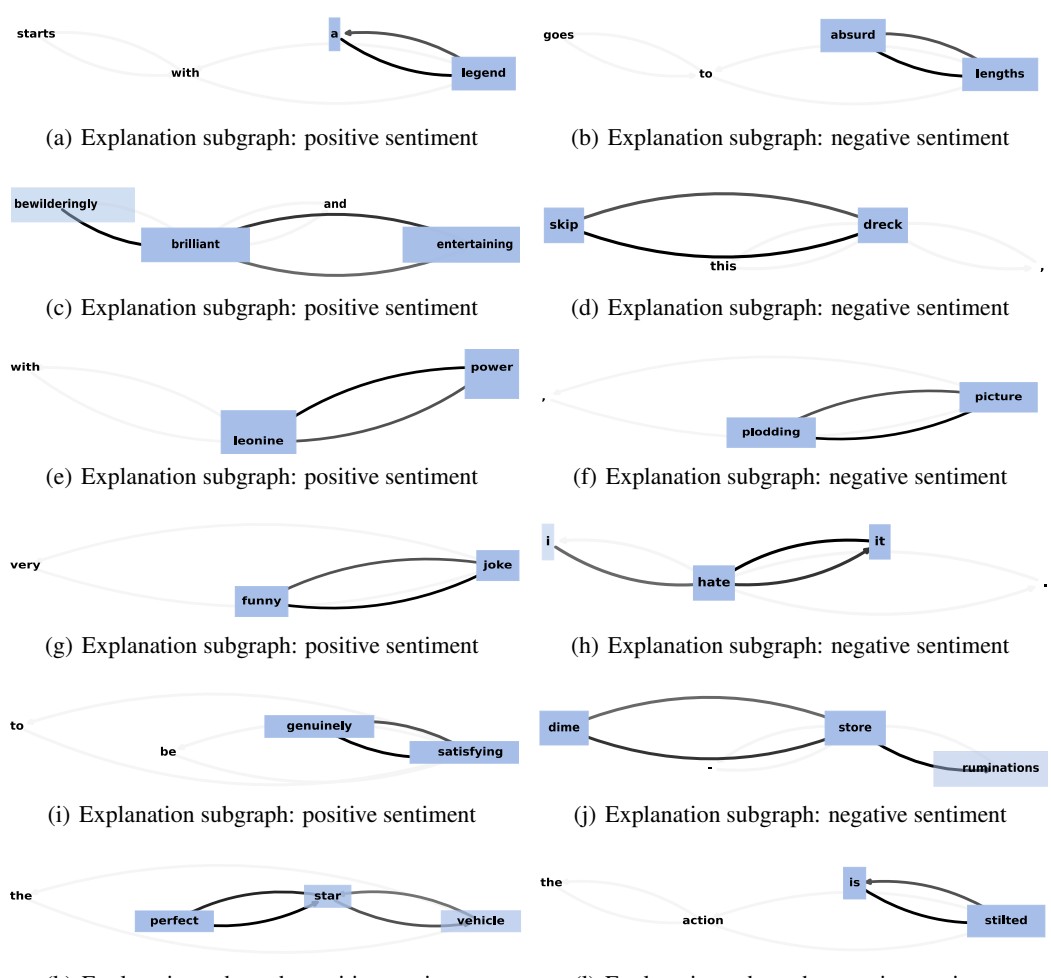

(a) Explanation subgraph: positive sentiment

(b) Explanation subgraph: negative sentiment

(c) Explanation subgraph: positive sentiment

(d) Explanation subgraph: negative sentiment

(e) Explanation subgraph: positive sentiment

(f) Explanation subgraph: negative sentiment

(g) Explanation subgraph: positive sentiment

(h) Explanation subgraph: negative sentiment

(i) Explanation subgraph: positive sentiment

(j) Explanation subgraph: negative sentiment

(k) Explanation subgraph: positive sentiment

(l) Explanation subgraph: negative sentiment

Figure 7: Visualization of $\pi$-GNN explanation on the Graph-SST2 dataset. Each graph represents a comment, where the explanations are highlighted by bule boxes and bolder lines.

Table 10: The impact of pre-training datasets with different sizes.

| Task | Dataset | PT-Motifs | W/O Pre-train | PT-Motifs-M | PT-Motifs-S |
|---|---|---|---|---|---|
| Interpretation | BA-2Motifs | 99.33 | 93.19 | 98.91 | 97.07 |
| | Mutag | 99.81 | 95.29 | 99.06 | 96.28 |
| Prediction | Ogbg-Molhiv | 80.86 | 79.71 | 80.77 | 79.82 |
| | Graph-SST2 | 88.05 | 83.48 | 87.59 | 85.23 |

explanatory $G_e$ from uniform distribution, while the base $G_b$ is determined by the following formula,

$$P(G_b) = b \times I(G_b = G_e) + \frac{1-b}{4} \times I(G_b \neq G_e). \tag{17}$$

Therefore, we can manipulate b to control the imbalance degree and the imbalance degree is defined as $i = 4b/(1-b)$. The corresponding result is reported in Table 11. The results demonstrate that when pretraining on an imbalanced dataset, the performance improvement is less significant than that on the balanced one, but still better than that without pretraining. Moreover, an overly imbalanced pre-training dataset ($b = 0.7, i \approx 9.3$) may cause negative transfer issue (on Muatg, Molhiv, and Graph-SST2). Therefore, the pre-training dataset ought to be balanced and thus can mitigate the negative transfer issue to some extents.

Table 11: The impact of pre-training datasets with different sizes.

| Task | Dataset | PT-Motifs | W/O Pre-train | PT-Motifs-M | PT-Motifs-S |
|---|---|---|---|---|---|
| Interpretation | BA-2Motifs | 99.33 | 93.19 | 98.91 | 97.07 |
| | Mutag | 99.81 | 95.29 | 99.06 | 96.28 |
| Prediction | Ogbg-Molhiv | 80.86 | 79.71 | 80.77 | 79.82 |
| | Graph-SST2 | 88.05 | 83.48 | 87.59 | 85.23 |

# F  Hyper-parameter Analysis

In Figure 8, we conduct analysis on the number of basic pattern-learners, to evaluate the tendency of $\pi$-GNN performance with the increase of the pattern-learner number. Additionally, we report the performance of the variant without pre-training phase, which is marked by the subscript "DFT". One can observe that as the number of basic learners increasing, both the interpretation and prediction performance generally improves, which demonstrates the co-existence of multiple structural patterns. For the Mutag dataset, when we increase the number to 2, the performance improvement (5.76%) is the most significant. But when the number increases to 16 and 32, the performance is inferior to that of 8 basic pattern-learners. We ascribe this degradation to the difficulty of integrating more and more patterns when the basic learner number increasing. For the Graph-SST2 dataset, the prediction performance consistently improves along with the number of basic learners. This may indicate that the structural patterns in Graph-SST2 are more intricate than those in Mutag dataset. Moreover, the induced variant without the pre-training phase is superior to the $\pi$-GNN model consistently, which again verifies the effectiveness of the explainer pre-training phase, by 3.32% ROC-AUC score on Mutag dataset and 4.08% accuracy on Graph-STT2 dataset on average.

# G  Derivation

First, we restate and prove **Theorem 1** [44].

**Theorem 1.** *Let $\Sigma_n$ be the set of all adjacency matrix $\mathbf{A}$ with $n$ nodes. Given a graph $G = (\mathcal{V}, \mathcal{E}) \in \Sigma_n, n \geq 2$, let $\Gamma^*(S, \mathbf{A})$ be a most-expressive structural representation of nodes set $S \subseteq \mathcal{V}$ in $G$. $\forall \mathbf{A} \in \Sigma_n$, there exists a most-expressive node representation $\mathbf{Z}^*|\mathbf{A}$ satisfies the relationship as follows,*

$$\Gamma^*(S, \mathbf{A}) = \mathbb{E}_{\mathbf{Z}^*}[f^{(|S|)}((\mathbf{Z}_v^*)_{v \in S})|\mathbf{A}], \forall S \subseteq \mathcal{V}, \tag{18}$$

*for an appropriate $k$-variable function $f^{(k)}(\cdot)$.*

*Proof.* Given graph $G = (\mathcal{V}, \mathcal{E})$ with $n$ nodes, we construct an equivalent set of the most-expressive structural representation $\Gamma^*(S, \mathbf{A})$, with permutations on node indices:

$$\Pi(\mathbf{A}) = \left\{ \Gamma^*\Big(v, \mathbf{A}, \pi(1, 2, \cdots, n)\Big)_{\forall v \in \mathcal{V}} \Big| \pi \in \Pi_n \right\} \tag{19}$$

Define $\mathbf{Z}^*|\mathbf{A}$ as the random variable with a uniform measure over $\Pi(\mathbf{A})$. Assume the node subset $S$ has no other joint isomorphic set $S'$. Then, for any such $S$ and any element $\gamma_\pi \in \Pi(\mathbf{A})$ as follow,

$$\gamma_\pi = \Gamma^*\Big(v, \mathbf{A}, \pi(1, 2, \cdots, n)\Big)_{\forall v \in \mathcal{V}}, \tag{20}$$

there exists a bijective measurable map between the nodes in $S$ and their positions in the representation vector $\gamma_\pi$. Next, we consider the representation set $\mathcal{O}_S(\mathbf{A})$ restricted to the node subset $S$ as follows,

$$\mathcal{O}_S(\mathbf{A}) := \left\{ \Gamma^*\Big(v, \mathbf{A}, \pi(1, 2, \cdots, n)\Big)_{\forall v \in S} \Big| \pi \in \Pi_n \right\} = \left\{ \Big((\mathbf{Z}_v^*)_{v \in S}|\mathbf{A}\Big) \right\}, \tag{21}$$

and prove that there exists an surjection between $\mathcal{O}_S(\mathbf{A})$ and $\Gamma^*(S, \mathbf{A})$. The surjection exists if, $\forall$ non-isomorphic node subset $S_1, S_2$, it implies $\mathcal{O}_{S_1}(\mathbf{A}) \neq \mathcal{O}_{S_2}(\mathbf{A})$. This condition naturally holds if $|S_1| \neq |S_2|$. When $|S_1| = |S_2|$, we prove by contradiction and assume $\mathcal{O}_{S_1}(\mathbf{A}) = \mathcal{O}_{S_2}(\mathbf{A})$. Since the node indices is unique and $\Gamma^*$ is most-expressive, the representation $\Gamma^*(v, \mathbf{A}, \pi(1, 2, \cdots, n))$ of node $v$ and permutation $\pi$ is unique too. As $S_1$ is non-isomorphic to $S_2$, there must exist at least one node $u \in S_1$ that has no isomorphic equivalent element in $S_2$. Hence, $\exists \pi \in \Pi_n$ that provides a representation $\Gamma^*(u, \mathbf{A}, \pi'(1, 2, \cdots, n))$ and $\nexists \pi' \in \Pi_n, v \in S_2$ that the coresponding representation

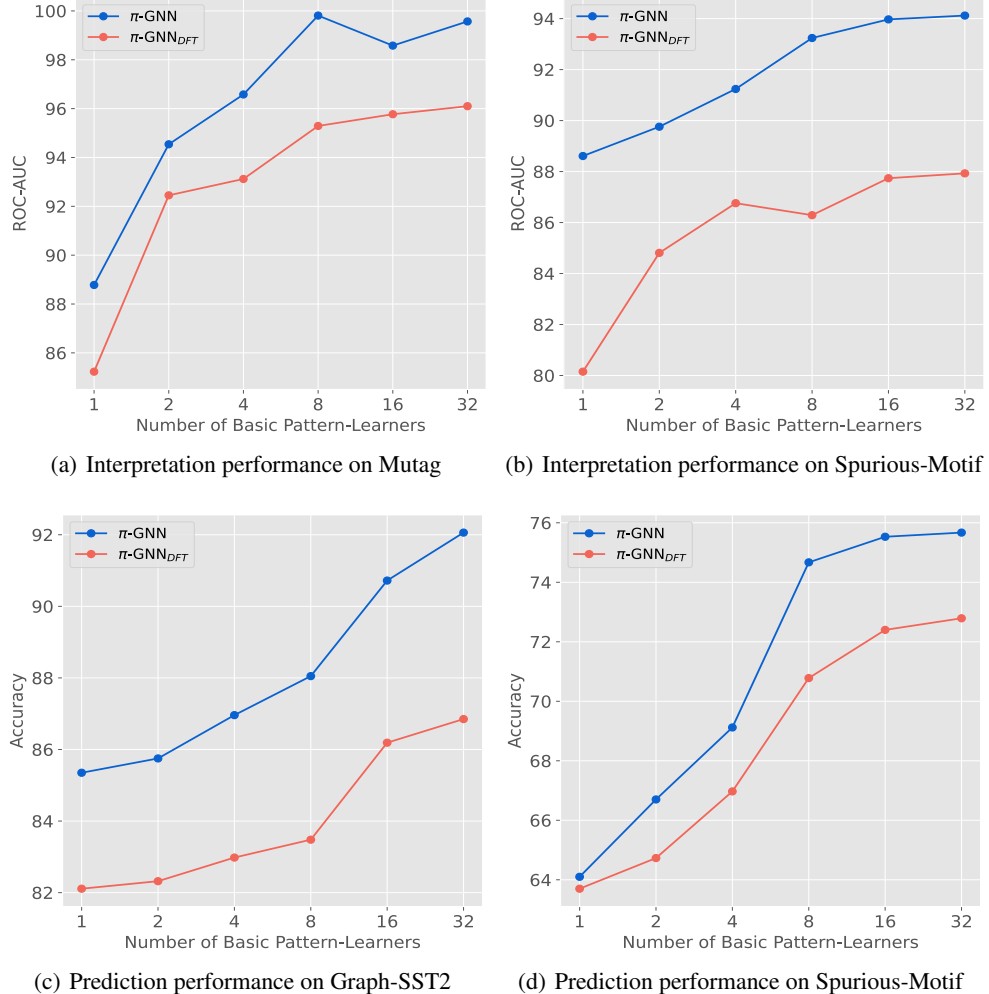

(a) Interpretation performance on Mutag

(b) Interpretation performance on Spurious-Motif

(c) Prediction performance on Graph-SST2

(d) Prediction performance on Spurious-Motif

Figure 8: The hyper-parameter analysis on the number of the basic pattern-learners.

$\Gamma^*(u, \mathbf{A}, \pi'(1, 2, \cdots, n))$ matches. Finally, we conclude a contradiction from the original assumption $\mathcal{O}_{S_1}(\mathbf{A}) = \mathcal{O}_{S_2}(\mathbf{A})$ and we know the surjection between $\mathcal{O}_S(\mathbf{A})$ and $\Gamma^*(S, \mathbf{A})$ does exist.

Furthermore, it has been proved that for finite multisets with real number elements, a most-expressive multiset function can be defined as the expectation of a function $f^{(|S|)}$ over the multiset. Therefore, there exists some subjective function $f^{(|S|)}$ whose expectation over $\mathcal{O}_S(\mathbf{A})$ gives $\Gamma^*(S, \mathbf{A})$.

Next, we restate and prove **Theorem 2**.

**Theorem 2.** *The structural representation of edge* $e = (v_i, v_j)$ *can be learnt by simply approaching a function* $f^{(2)}$ *which satisfies* $\Gamma(e, \mathbf{A}) = f^{(2)}(\mathbb{E}[(\mathbf{Z}_v)_{v \in \{i,j\}} | \mathbf{A}])$.

*Proof.* According to Theorem 1, $(\mathbf{Z}_v)_{v \in \{i,j\}} | \mathbf{A}$ can be represented as follows,

$$(\mathbf{Z}_v)_{v \in \{i,j\}} | \mathbf{A} = \varphi \Big( \Gamma(v, \mathbf{A})_{v \in \{i,j\}}, \epsilon_{\{i,j\}} \Big), \tag{22}$$

where the noise $\epsilon_{\{i,j\}}$ is marginalized from an independent noise distribution. With an assumption of $f^{(2)}$ that is able to capture the structural dependencies within the adjacent matrix $\mathbf{A}$, we can compute the expectation of $(\mathbf{Z}_v)_{v \in \{i,j\}} | \mathbf{A}$ and eliminate the noise.

# H   Algorithm

We present the algorithm of graph-hypergraph transformation as following.

---

**Algorithm 1** Graph-Hypergraph Transformation

---

**Input:** Edge index $\Omega \in \mathbb{R}^{2 \times |\mathcal{E}|}$ of raw graph $G$
**Output:** Hyperedge index $\Omega_h$ of hypergraph $G_h$
 1: **initialize** *hyperedge set* $\mathcal{E}_h \Leftarrow \emptyset$
 2: **initialize** *hyperedge index* $\Omega_h \Leftarrow \emptyset$
 3: **for** edge $e_i = (u, v)$ in $\Omega$ **do**
 4:     $\mathcal{E}_h^u \Leftarrow \mathcal{E}_h^u \cup \{i\}$, $\mathcal{E}_h^v \Leftarrow \mathcal{E}_h^v \cup \{i\}$ // Record the endpoints of each edge
 5: **end for**
 6: **for** the $p$-th hyperedge $\mathcal{E}_h^p$ in $\mathcal{E}_h$ **do**
 7:     **for** item $q$ in $\mathcal{E}_h^p$ **do**
 8:         $\Omega_h \Leftarrow \Omega_h \cup \{(p, q)\}$ // Allocate the same index $p$ for each individual hypernode $q$
 9:     **end for**
10: **end for**
11: **return** $\Omega_h$

---

We also provide an illustrative example of the transformation process in Figure 2 of Section 3.1.

# I   Limitations

At last, we provide open discussion about the limitations of the $\pi$-GNN model.

**Efficiency.** If the number of basic pattern-learners need to be increased to a large amount for some intricate graph datasets, the computational efficiency of $\pi$-GNN will become the bottleneck. Although the efficiency can be improved by introducing multi-thread computation, the time complexity of $\pi$-GNN with multiple basic pattern-learners is higher than the current interpretable GNNs.

**Feature Dimension.** The feature dimensions of the downstream graph datasets are usually different from that of the PT-Motifs dataset in pre-training phase. Therefore, we have to additionally introduce a linear layer to align the dimension differences between PT-Motifs and the downstream datasets. Though this linear layer can be merged into the hypergraph refining module, it indeed increases the optimization difficulty when fine-tuning.

**Node Feature.** $\pi$-GNN focuses on identifying the influential subgraphs by computing the edge contribution score, but it is unable to select the important fraction of the node features that leads to the model prediction. As a future direction, we consider to systematically extend $\pi$-GNN to the interpretation problem in node classification and link prediction task, where the node features are more informative and influential than those in the graph classification task.

**Unseen Structural Patterns.** As shown in Appendix B.2, several structural patterns are universal and generalize to the synthetic PT-Motifs dataset and the real-world datasets (e.g., Mutag, MNIST-75sp, Molhiv, Graph-SST2). During the pre-training phase over PT-Motifs dataset, $\pi$-GNN extracts these universal structural patterns and combines them with the local structural interactions to achieve generalizable interpretation. But for some unseen structural patterns that does not exist in PT-Motifs, $\pi$-GNN is unable to capture and then employ them to identify explanations.

