# OpenReview forum: "Train Once and Explain Everywhere: Pre-training Interpretable Graph Neural Networks"
_NeurIPS.cc/2023/Conference — NeurIPS 2023 poster_

### Official Review · Reviewer_34Ne · 2023-07-03

**Soundness:** 3 good
**Presentation:** 3 good
**Contribution:** 4 excellent
**Rating:** 8
**Confidence:** 4

**Summary:**

This paper proposed a novel pre-training method for interpretable graph neural network. Interpretable GNN is currently an important research issue and has attracted rising research attention recently. However, exiting methods are generally designed for some special types of datasets, and thus are hard to generalize well to other graphs or tasks. This paper makes the first attempt to design a pre-training framework with a carefully developed labeled synthetic graph dataset. As the synthetic graphs have ground truth explanations, the pretained model can better capture the common structure knowledge of different types of graphs. A structure pattern learning module and a hypergraph refining module are also proposed to make the pretained model achieve better performance. Generally, the paper is very well written and has clear contributions. It is a nice try to design pre-trained interpretable GNNs. The experiment is extensively studied over multiple datasets, and the performance improvement over current SOTA is significant. The evaluation on node classification and graph classification tasks further verifies the promising generalization ability of the model.

**Strengths:**

(1)	This paper for the first time proposed a pre-training interpretable GNN model, which is novel and practically important. The pre-trained model is first trained over a large synthetic graph datasets, and then fine-tuned over downstream graphs, which achieves significant performance improvement. As a general and pioneer pipeline, I believe the paper can potentially motivate a lot of following works.
(2)	A structure pattern learning module and a hypergraph refining module are proposed and integrated into the pre-training model. The two modules can better and more comprehensively capture the structural patterns and the edge interactions of the graphs.
(3)	The experiment is extensively studied over both real and synthetic graph datasets. The results show the significant performance improvement over current SOTA. The experiment is convincing and the code is publicly available.
(4)	The paper is very well written and organized in good logic. It is easy to read and follow.


**Weaknesses:**

(1)	As the constructed synthetic graphs with ground truth labels may significantly affect the performance of the pre-training model. How to construct a good synthetic graphs should be clearly explained. Currently, it is not clear enough on how to construct the data. More details on the synthetic datasets construction should be provided.
(2)	It is not very clear to me why the authors need to give Theorem 1 and 2. Based on my understanding, the authors want to prove that the learning edge representation can fully preserve the graph pattern information. If it is the case, the authors should explain it more clearly and describe why it is important for interpretable GNN.
(3)	In the evaluation, pi-GNN is also compared with pi-GNN-DFT, the authors explained that pi-GNN-DFT directly fine-tunes on downstream datasets without pre-training. How to implement that exactly is not explained.


**Questions:**

(1)	How to construct a good synthetic graph datasets for pretaining? Are the ground truth GNN explanations labeled manually?
(2)	In section 4.3, the authors evaluate the model pre-trained over graph classification dataset on the explanation task of node classification to evaluate the inter-task generation performance. How to do that exactly? The learned graph embeddings learned from graph classification is used for node classification?


**Limitations:**

The authors have adequately addressed the limitations of their work in the appendix.

---

> ### Author Rebuttal · Authors · 2023-08-10
>
> >Q1: As the constructed synthetic graphs with ground truth labels may significantly affect the performance of the pre-training model. How to construct good synthetic graphs should be clearly explained. Currently, it is not clear enough on how to construct the data. More details on the synthetic dataset construction should be provided. How to construct a good synthetic graph dataset for pretraining? Are the ground truth GNN explanations labeled manually?
>
> A1: Thanks for your suggestion.
>
> 1). We will add the generation details of the synthetic dataset in Appendix. Following existing works [1, 2], each synthetic graph consists of a base and an influential motif and the label is determined by the motif solely. Given one certain motif, we first sample a base from a uniform distribution over all bases and the motif is attached a randomly selected node in the base.
>
> 2). According to our experimental results, a good synthetic graph dataset for pre-training should first be balanced. Moreover, larger dataset size and more motif classes are also promising to guarantee the effectiveness of pre-training process.
> >Q2: It is not very clear to me why the authors need to give Theorem 1 and 2. Based on my understanding, the authors want to prove that the learning edge representation can fully preserve the graph pattern information. If it is the case, the authors should explain it more clearly and describe why it is important for interpretable GNN.
>
> A2: Thanks for your comments. Your understanding is absolutely correct and we will add more explanation about the importance of the edge representation with regard to interpretable GNNs. The edges are more essential to GNN explanation compared with nodes, as the previous works [3, 4] point out. Therefore, we need a more expressive representation of edges and thus can provide a better GNN explanation.
> >Q3: In the evaluation, pi-GNN is also compared with pi-GNN-DFT, the authors explained that pi-GNN-DFT directly fine-tunes on downstream datasets without pre-training. How to implement that exactly is not explained.
>
> A3: Thanks for your suggestion. We will append more details of the directly fine-tuning paradigm in the final version. Specifically, pi-GNN-DFT skips the pretraining process on the synthetic dataset and conduct standard training on the downstream datasets just like the other baselines.
> >Q4: In section 4.3, the authors evaluate the model pre-trained over graph classification dataset on the explanation task of node classification to evaluate the inter-task generation performance. How to do that exactly? The learned graph embeddings learned from graph classification is used for node classification?
>
> A4: Thanks for your comment. We will append more details of the inter-task evaluation in the final version. When evaluating the inter-task interpretation performance, the whole graph of the node classification task is fed into the pretrained explainer and the explainer will identify the explanatory subgraph of each target node in test set. The interpretation performance is evaluated on the target nodes.
>
> [1] GNNExplainer: Generating Explanations for Graph Neural Networks. NeurIPS 2019.
>
> [2] Graph Information Bottleneck for Subgraph Recognition. ICLR 2021.
>
> [3] Discovering Invariant Rationales for Graph Neural Networks. ICLR 2022.
>
> [4] Parameterized Explainer for Graph Neural Network. NeurIPS 2020.

---

> ### Author Response · Authors · 2023-08-20
>
> Dear reviewer 34Ne:
>
> Thanks again for your insightful comments, which, we believe, are very important to improve our paper.
>
> In the rebuttal and submitted one-page pdf, we have tried to answer your questions one by one.
>
> If you have further questions, we are very happy to discuss them.

---

> > ### Comment · Reviewer_34Ne · 2023-08-20
> > **Response to rebuttal**
> >
> > The authors have addressed my concerns in the rebuttal, and I will keep my socre as clear accept.

---

> > > ### Author Response · Authors · 2023-08-21
> > >
> > > Thanks again for your positive opinion of our paper and your valuable comments.

---

### Official Review · Reviewer_XLHb · 2023-07-06

**Soundness:** 3 good
**Presentation:** 3 good
**Contribution:** 3 good
**Rating:** 6
**Confidence:** 4

**Summary:**

This paper presents a novel method for GNN explainability. The key innovation of this method is that of relying on synthetic graphs with known explanations to pretrain the model. The pretraining helps to learn general explainability patterns, introducing an inductive bias. Such patterns are then aggregated and refined through specific modules. The method is experimentally evaluated on synthetic and real-world graphs, and compared to different methods recently published in the literature.

**Strengths:**

- The manuscript is well-written and relatively easy to follow.
- The empirical performances of the reported experiments are very compelling.

**Weaknesses:**

- The main weakness in my opinion is that the dependency between the test tasks and the synthetic pre-training dataset is not well investigated. However, the synthetic dataset is probably one of the most important hyperparameters, which could be difficult to optimize in rael-world settings. I think that the authors should evaluate the impact of different pretraining datasets on different test tasks, also making sure that some test tasks include motifs not fully covered in the pretraining tasks.

**Questions:**

See previous point.

**Limitations:**

Limitations are well addressed.

---

> ### Author Rebuttal · Authors · 2023-08-10
>
> >Q1: The main weakness in my opinion is that the dependency between the test tasks and the synthetic pre-training dataset is not well investigated. However, the synthetic dataset is probably one of the most important hyper-parameters, which could be difficult to optimize in real-world settings. I think that the authors should evaluate the impact of different pre-training datasets on different test tasks, also making sure that some test tasks include motifs not fully covered in the pre-training tasks.
>
> A1: Thanks for your suggestion.
>
> 1). It is true that the synthetic pretraining dataset is of great importance to the downstream tasks and how to select a good pre-training dataset may be difficult in real-world settings. To further evaluate the impact of different pretraining datasets on different test tasks, we have added extended experiments on pre-training datasets with different classes of motifs. As we state, the PT-Motifs dataset contains 5 motifs (i.e., Diamond, House, Crane, Cycle, and Star). To investigate the impact of motifs in pre-training dataset, we generate two dataset that contains only 3 motifs. Specifically, PT-DCrS contains Diamond, Crane, and Star and PT-DCyH contains Diamond, Cycle and House. Note that the House and Cycle motifs in BA-2Motifs are not covered by PT-DCrS. The result is reported in the following table.
> |Interpretation|BA-2Motifs|Mutag|
> |-|-|-|
> |PT-Motifs|99.33|99.81|
> |w/o pre-train|93.19|95.29|
> |PT-DCrS|95.53|98.56|
> |PT-DCyH|99.04|98.31|
>
> |Prediction|Molhiv|Graph-SST2|
> |-|-|-|
> |PT-Motifs|80.86|88.05|
> |w/o pre-train|79.71|83.48|
> |PT-DCrS|79.95|87.26|
> |PT-DCrS|79.98|88.02|
>
> The results demonstrate that even when the pre-training dataset does not cover all the motifs in downstream datasets, the pre-training process can still improve the interpretation and prediction performance. Moreover, when fine-tuning on BA-2Motifs, pre-training on PT-DCyH significantly outperforms that on PT-DCrS.
>
> 2). Moreover, we investigate the impact of imbalanced datasets. Following existing work [3], each synthetic graph consists of one base subgraph $G_b$ and one explanatory subgraph $G_e$. To generate the imbalanced datasets, we sample the explanatory $G_e$ from uniform distribution, while the base $G_b$ is determined by $P(G_b) = b\times I(G_b=G_e) + (1-b)/4\times I(G_b\neq G_e)$. Therefore, we can manipulate the hyperparameter b to control the imbalance degree and the imbalance degree is defined as $i=4b/(1-b)$. The experimental results of imbalanced pre-training datasets are listed in the following tables.
> |Interpretation|BA-2Motifs|Mutag|
> |-|-|-|
> |Balanced|99.33|99.81|
> |w/o pre-train|93.19|95.29|
> |b=0.3, i≈1.7|98.91|98.46|
> |b=0.5, i=4.0|96.15|97.10|
> |b=0.7, i≈9.3|93.63|94.30|
>
> |Prediction|Molhiv|Graph-SST2|
> |-|-|-|
> |Balanced|80.86|88.05|
> |w/o pre-train|79.71|83.48|
> |b=0.3, i≈1.7|79.42|85.34|
> |b=0.5, i=4.0|77.18|83.14|
> |b=0.7, i≈9.3|75.04|81.90|
>
> The results demonstrate that when pretraining on an imbalanced dataset, the performance improvement is less significant than that on the balanced one, but still better than that without pretraining.
>
> 3). Compared with PT-Motifs (80000 graphs) in our paper, we generate PT-Motifs-M with 50000 graphs and PT-Motifs-S with 10000 graphs. The experimental results on the pretraining datasets with different scales are listed in the following table.
> |Interpretation|BA-2Motifs|Mutag|
> |-|-|-|
> |PT-Motifs (80,000)|99.33|99.81|
> |w/o pre-train|93.19|95.29|
> |PT-Motifs-M (50,000)|98.91|99.06|
> |PT-Motifs-S (10,000)|97.07|96.28|
>
> |Prediction|Molhiv|Graph-SST2|
> |-|-|-|
> |PT-Motifs (80,000)|80.86|88.05|
> |w/o pre-train|79.71|83.48|
> |PT-Motifs-M (50,000)|80.77|87.59|
> |PT-Motifs-S (10,000)|79.82|85.23|
>
> The results show that even the PT-Motifs-S is able to outperform the model without pre-training. Moreover, we can notice that the large pre-training dataset can indeed improve the performance more significantly than that with small size.
>
> [1] Discovering Invariant Rationales for Graph Neural Networks. ICLR 2022.
>
> [2] Parameterized Explainer for Graph Neural Network. NeurIPS 2020.

---

> > ### Comment · Reviewer_XLHb · 2023-08-16
> > **Response to rebuttal**
> >
> > Thanks for the response and for the careful analysis, which helps elucidate the connection between test tasks and pretraining datasets. This was my main concern reading the paper. I confirm my acceptance score of 6.

---

> > > ### Author Response · Authors · 2023-08-16
> > >
> > > Thanks again for your careful reading, valuable comments, and constructive suggestions, which have significantly improved our manuscript.

---

### Official Review · Reviewer_M6tM · 2023-07-06

**Soundness:** 3 good
**Presentation:** 3 good
**Contribution:** 2 fair
**Rating:** 5
**Confidence:** 4

**Summary:**

The paper proposes a generalizable GNN interpretation model, aiming to learn the universal structural patterns of graphs so that it can be applied any downstream applications.

**Strengths:**

(1)	The problem that the paper studies is very interesting, a model trained to identify the universal explanatory subgraph in different cases will be of great practical use.
(2)	Overall, the paper is easy to follow.
(3) The authors provide theoretical analysis to support their claim.

**Weaknesses:**

(1)	Although the paper provides empirical study to demonstrate the effectiveness of the hypergraph refining module, the motivation to incorporate such a component is still not very clear. What is the advantage of the hypergraph refining module compared with a normal graph?
(2)	Since the explanation model is pre-trained, the authors are encouraged to incorporate some graph SSL methods as baselines for fair comparison (like GraphCL and GraphLoG).
(3)	It seems the synthetic pretraining dataset is of great importance to the final results, therefore more empirical studies should the conducted to investigate the impacts of the pretraining dataset on the final results, like the dataset size and distribution. What if the synthetic dataset is imbalanced and what if the distribution of the pretraining dataset and downstream datasets are not aligned? Also, I wonder about the cost to generate such a synthetic dataset.
(4)	The authors are encouraged to provide the training details of the baselines.

**Questions:**

Refer the weakness section

---

> ### Author Rebuttal · Authors · 2023-08-10
>
> >Q1: The motivation of the hypergraph refining module and its advantages.
>
> A1: For GNN explainers, the edges and the edge interactions are more essential compared with nodes and node interactions [1,2]. Therefore, we need a more expressive edge representation learning paradigm. To capture the edge interactions, we propose to exchange the roles of edge and node, and then perform the message passing mechanism. In this way, the edges can interact with each other via message passing and the interactions among edges can be captured by the learned representations. However, after exchanging roles, each edge may connect with several edges via a single node, where the normal graph neural networks cannot be directly applied. Hence, we propose the hypergraph refining module to implement the expressive learning on edge representations, where the edges are converted to hyper-nodes and the nodes become hyper-edges. The ablation study shows that, compared with normal GNN models, adopting the proposed hypergraph refining module improves the GNN explanation performance from 94.42% to 99.81% on Mutag dataset, which verifies the effectiveness of the hypergraph refining module.
> >Q2: Incorporate some graph SSL methods as baselines for fair comparison.
>
> A2: We subscribe the graph encoder in our model with a 2-layer GIN which is pretrained by GraphCL [3] and InfoGraph [4]. **The results are listed in Table 1 of the attached PDF.** As the results show, the graph SSL pretrained explainers are inferior to the proposed model after pretraining. As we surmise, it's because the graph SSL methods are general pre-training paradigm for GNNs, instead of specifically designing for the GNN interpretation problem.
> >Q3: More empirical studies to investigate the impacts of the pretraining dataset.
>
> A3: 1). To further investigate the impact of pre-training dataset size, we have added supplement experiments. Specifically, compared with PT-Motifs (80000 graphs) in our paper, we generate PT-Motifs-M with 50000 graphs and PT-Motifs-S with 10000 graphs. The results are listed as follows.
> |Interpretation|BA-2Motifs|Mutag|
> |-|-|-|
> |PT-Motifs (80000)|99.33|99.81|
> |w/o pretrain|93.19|95.29|
> |PT-Motifs-M (50000)|98.91|99.06|
> |PT-Motifs-S (10000)|97.07|96.28|
>
> |Prediction|Molhiv|Graph-SST2|
> |-|-|-|
> |PT-Motifs (80000)|80.86|88.05|
> |w/o pretrain|79.71|83.48|
> |PT-Motifs-M (50000)|80.77|87.59|
> |PT-Motifs-S (10000)|79.82|85.23|
>
> The results show that even the PT-Motifs-S is able to outperform the model without pre-training. Moreover, we can notice that a large pre-training dataset can indeed improve the performance more significantly than that with small size.
> 2). It is possible that if the synthetic pre-training dataset is imbalanced, the performance improvement on downstream tasks will degrade, in terms of both the interpretation and prediction. To further investigate this issue, we have added supplement experiments on imbalanced pre-training dataset. Following existing works [5,6], to generate the imbalanced datasets, we sample the explanatory $G_e$ from uniform distribution, while the base $G_b$ is determined by $P(G_b) = b\times I(G_b=G_e) + (1-b)/4\times I(G_b\neq G_e)$. Therefore, we can manipulate b to control the imbalance degree and the imbalance degree is defined as $i=4b/(1-b)$. The results are listed as follows.
> |Interpretation|BA-2Motifs|Mutag|
> |-|-|-|
> |Balanced|99.33|99.81|
> |w/o pretrain|93.19|95.29|
> |b=0.3, i≈1.7|98.91|98.46|
> |b=0.5, i=4.0|96.15|97.10|
> |b=0.7, i≈9.3|93.63|94.30|
>
> |Prediction|Molhiv|Graph-SST2|
> |-|-|-|
> |Balanced|80.86|88.05|
> |w/o pretrain|79.71|83.48|
> |b=0.3, i≈1.7|79.42|85.34|
> |b=0.5, i=4.0|77.18|83.14|
> |b=0.7, i≈9.3|75.04|81.90|
>
> The results demonstrate that when pretraining on an imbalanced dataset, the performance improvement is less significant than that on the balanced one, but still better than that without pretraining.
> 3). In fact, the distribution of our pretraining dataset is not always aligned with the downstream tasks. Generally, the pretraining process is not strongly coupling with the downstream tasks as in CV [5] and NLP [6]. Hence, there is no need to deliberately align the pretraining dataset with the downstream datasets. We need the fine-tuning process to align the pre-trained model with the specific downstream tasks. Thus, the quantity and quality of downstream dataset are important to the fine-tuned performance too. Additionally, when the distribution of our pretraining dataset is severely not aligned with the downstream datasets and causes the negative transfer issue, we can adjust the synthetic algorithm to align the downstream dataset, which is the main advantage of synthetic pretraining paradigm.
>
> 4). **We list the time cost to generate a synthetic graph dataset with different graph scales in Table 2 of the attached PDF.** The result shows that the time cost of graph generation is nearly a linear function of the average node numbers.
> >Q4: The training details of the baselines.
>
> A4: Here, we sketch two important baselines. For GNNExplainer, we tune the learning rate from (1,0.1,0.01,0.001) and the coefficient of the L1-norm from (0.1,0.01,0.001). The coefficient of the entropy regularization is set to the recommended value 1. For PGExplainer, we use the tuned recommended settings from [6], including the temperature, the coefficient of L1-norm regularization. The training details of the baselines will be added in the final version.
>
> [1] Discovering Invariant Rationales for Graph Neural Networks. ICLR 2022.
>
> [2] Parameterized Explainer for Graph Neural Network. NeurIPS 2020.
>
> [3] Graph Contrastive Learning with Augmentations. NeurIPS 2020.
>
> [4] InfoGraph: Unsupervised and Semi-supervised Graph-Level Representation Learning via Mutual Information Maximization. ICLR 2020.
>
> [5] Momentum contrast for unsupervised visual representation learning. CVPR 2020.
>
> [6] BERT: Pre-training of Deep Bidirectional Transformers for Language Understanding. NAACL-HLT 2019.

---

> ### Author Response · Authors · 2023-08-20
>
> Dear reviewer M6tM:
>
> Thanks again for your insightful comments, which, we believe, are very important to improve our paper.
>
> In the rebuttal and submitted one-page pdf, we have tried to answer your questions one by one.
>
> If you have further questions, we are very happy to discuss them.

---

> > ### Comment · Reviewer_M6tM · 2023-08-21
> > **Thanks for your response**
> >
> > Thanks for the detailed response provided by authors and I want to express my appreciation for the additional illustrations and experiments. I decide to raise my rating to 5 (borderline accept) and make the final decision until the reviewer discussion phase. Thanks!

---

> > > ### Author Response · Authors · 2023-08-21
> > >
> > > Thanks again for your valuable comments and insightful suggestions that have allowed us to improve the manuscript.

---

### Official Review · Reviewer_EENr · 2023-07-07

**Soundness:** 3 good
**Presentation:** 4 excellent
**Contribution:** 3 good
**Rating:** 8
**Confidence:** 4

**Summary:**

The authors propose a pre-trained interpretable GNN named \pi-GNN that can distill universal graph structural patterns. \pi-GNN is pre-trained on a newly constructed synthetic graph datasets with ground-truth explanations and then able to generalize across different graph datasets and tasks. Technically, a structural pattern learning module is introduced to capture and fuse multiple structural patterns for generalizable graph representations. Next, a refining module based on hypergraph is proposed to incorporate the generalizable patterns with the local structural interactions. Extensive experiments demonstrate the superiority of \pi-GNN over the SOTA baselines in terms of both interpretation and prediction performance. Additionally, an inter-task experiment which evaluates the graph classification pre-trained model on node classification task strongly verifies the excellent generalizability of \pi-GNN.

**Strengths:**

S1: The paper first studies the pre-training problem of interpretable GNN for generalizable graph interpretability, which is interesting and insightful to the community.
S2: The proposed method, i.e., \pi-GNN, is well-motivated and plausible, since the graph structure follows some universal structural patterns that is important to the graph interpretation problem. Specifically, the combination of multiple basic pattern learners and one integrated pattern learner is reasonable to provide a generalizable structural representation. The graph-to-hypergraph transformation is elegant to fuse the universal patterns and the local interactions via hypergraph message passing. In short, the architecture of \pi-GNN is easy to understand and the figures in the manuscript are clearly illustrated.
S3: The improvements shown in experiment are significant on almost all datasets with regard to both interpretation and prediction performance. In section 4.3, after pre-trained on graph classification task, the top-tier interpretation performance on node classification is astonishing and demonstrates the cross-task generalizability of \pi-GNN.
S4: Sufficient supplementaries are provided, including ablation study, hyper-parameter analysis, to probe into the effectiveness of the proposed modules and the suitable hyper-parameter of \pi-GNN. Moreover, a great deal of visualized explanation cases are reported for an intuitive understanding towards the interpretation and prediction of \pi-GNN.

**Weaknesses:**

W1: It seems like some related works about interpretable GNNs are missing in section 5, such as CAL [1] and OrphicX [2].
W2: The subgraph selection process is a little obscure to me.
The authors may want to append further illustrations on how to select the explanatory edges and embrace the contribution score into gradient optimization.
[1] Yongduo Sui, Xiang Wang, Jiancan Wu, Min Lin, Xiangnan He, Tat-Seng Chua. Causal Attention for Interpretable and Generalizable Graph Classification. KDD 2022: 1696-1705
[2] Wanyu Lin, Hao Lan, Hao Wang, Baochun Li. OrphicX: A Causality-Inspired Latent Variable Model for Interpreting Graph Neural Networks. CVPR 2022: 13719-13728

**Questions:**

Q1: The main question to me is the effectiveness of the pre-training strategy on interpretable GNNs. In what cases will this strategy improve the final interpretation and prediction performance? And how to avoid the negative transfer issue of the pre-training paradigm also needs more research efforts. Moreover, to consolidate the assumption that the universal patterns behind different tasks is common, the inter-task evaluation need to be conducted on some real-world datasets.
Q2: Will the SOTA baselines get improved by pre-training on the newly constructed datasets? There may need some modifications on the existing models to fit the pre-training paradigm, but I think it is profoundly influential to the community if the interpretable GNNs can be generally improved.

**Limitations:**

The authors have listed some limitations in the supplementary material. I still expect more exploration and discussion on the effectiveness of the pre-training strategy as I mentioned above.

---

> ### Author Rebuttal · Authors · 2023-08-10
>
> >Q1: It seems like some related works about interpretable GNNs are missing in section 5, such as CAL [1] and OrphicX [2].
>
> A1: Thanks for your suggestion. We will add the suggested references in the final version.
> >Q2: The subgraph selection process is a little obscure to me. The authors may want to append further illustrations on how to select the explanatory edges and embrace the contribution score into gradient optimization.
>
> A2: Thanks for your suggestions. Following existing works [3, 4, 5], the explainer will provide a mask matrix $M\in\mathbb{R}^{|V|\times|V|}$, where the element $M_{ij}$ indicates the importance of edge $(i,j)$. Afterwards, such a mask matrix can result in an attentive matrix $A^{att} = A\odot f(M)$, where $A$ is the adjacent matrix of the raw graph and the element $A^{att}_{ij}$ indicates the probability of edge $(i,j)$ belonging to the explanatory subgraph. Finally, based on the attentive matrix $A^{att}$, the explanatory graph is sampled according to edge probability. We will add more explanation about the subgraph selection process in the Appendix.
> >Q3: The main question to me is the effectiveness of the pre-training strategy on interpretable GNNs. In what cases will this strategy improve the final interpretation and prediction performance? And how to avoid the negative transfer issue of the pre-training paradigm also needs more research efforts. Moreover, to consolidate the assumption that the universal patterns behind different tasks is common, the inter-task evaluation need to be conducted on some real-world datasets.
>
> A3: Thanks for your comments and suggestions.
>
> 1). The pre-training strategy on the interpretable GNNs is significantly effective when the downstream datasets share some common structural patterns with the pre-training dataset. Therefore, a pre-training dataset that covers more structural patterns is more helpful to the downstream fine-tuning tasks.
>
> 2). As to the negative transfer issue, we notice that an imbalanced pre-training dataset tends to improve less significantly than a balanced dataset, in terms of both interpretation and prediction. We further investigate the impact of an imbalanced pre-training dataset. Following existing work [3, 4], each synthetic graph consists of one base subgraph $G_b$ and one explanatory subgraph $G_e$. To generate the imbalanced datasets, we sample the explanatory $G_e$ from uniform distribution, while the base $G_b$ is determined by $P(G_b) = b\times I(G_b=G_e) + (1-b)/4\times I(G_b\neq G_e)$. Therefore, we can manipulate the hyperparameter b to control the imbalance degree and the imbalance degree is defined as $i=4b/(1-b)$. The experimental results of imbalanced pre-training datasets are listed in the following tables.
> |Interpretation|BA-2Motifs|Mutag|
> |:-|:-|:-|
> |Balanced|99.33|99.81|
> |w/o pre-train|93.19|95.29|
> |b=0.3, i≈1.7|98.91|98.46|
> |b=0.5, i=4.0|96.15|97.10|
> |b=0.7, i≈9.3|93.63|94.30|
>
> |Prediction|Molhiv|Graph-SST2|
> |:-|:-|:-|
> |Balanced|80.86|88.05|
> |w/o pre-train|79.71|83.48|
> |b=0.3, i≈1.7|79.42|85.34|
> |b=0.5, i=4.0|77.18|83.14|
> |b=0.7, i≈9.3|75.04|81.90|
>
> The results reveal that an overly imbalanced pre-training dataset (b=0.7, i≈9.3) may cause negative transfer issue (on Muatg, Molhiv, and Graph-SST2). Therefore, the pre-training dataset ought to be balanced and thus can mitigate the negative transfer issue to some extents.
>
> 3). In this work, we just conduct some preliminary experiments to investigate the inter-task generalization problem and more evaluations on the real-world datasets are necessary to consolidate our assumption about universal patterns. In the future work, we will systematically extend to the interpretation problem in node classification and link prediction task.
> >Q4: Will the SOTA baselines get improved by pre-training on the newly constructed datasets? There may need some modifications on the existing models to fit the pre-training paradigm, but I think it is profoundly influential to the community if the interpretable GNNs can be generally improved.
>
> A4: Thanks for your comment. It is promising that the SOTA interpretable GNNs will get improved by incorporating the pre-training process. But on the other hand, incorporating the pre-training and fine-tuning paradigm with the interpretable GNNs is indeed challenging. For a future direction, we will investigate how to introduce the pre-training process to the SOTA baselines in a general way instead of specified modification.
>
> [1] Causal Attention for Interpretable and Generalizable Graph Classification. KDD 2022.
>
> [2] OrphicX: A Causality-Inspired Latent Variable Model for Interpreting Graph Neural Networks. CVPR 2022.
>
> [3] Parameterized Explainer for Graph Neural Network. NeurIPS 2020.
>
> [4] Towards Multi-Grained Explainability for Graph Neural Networks. NeurIPS 2021.
>
> [5] Interpretable and Generalizable Graph Learning via Stochastic Attention Mechanism. ICML 2022.

---

> ### Author Response · Authors · 2023-08-20
>
> Dear reviewer EENr:
>
> Thanks again for your insightful comments, which, we believe, are very important to improve our paper.
>
> In the rebuttal and submitted one-page pdf, we have tried to answer your questions one by one.
>
> If you have further questions, we are very happy to discuss them.

---

> > ### Comment · Reviewer_EENr · 2023-08-21
> > **Thanks for the rebuttal**
> >
> > Thanks for the rebuttal and detailed responses of the authors. I have carefully read the responses to my previous concerns. Generally, I am satisfied with their responses as most of my previous concerns, such as further explanation on the subgraph selection process, more discussions on the effectiveness evaluation of the model, and the generalization ability of the model are well addressed.
> > I think this paper makes clear contributions as this is the first try to build a more general pre-training GNN explainer. I believe this will motivate a lot of the following works.

---

> > > ### Author Response · Authors · 2023-08-21
> > >
> > > Thanks again for your positive opinion of our paper and your valuable comments.

---

### Author Rebuttal · Authors · 2023-08-10

We thank all the reviewers for their insightful and constructive feedback. We have made point-to-point response to the comments of each reviewer.

Moreover, we report two supplemental experiments in the attached file.

Finally, we once again thank all reviewers for their insightful comments which are very helpful for improving the quality of our paper.

---

### Decision · Program_Chairs · 2023-09-21

**Decision:**

Accept (poster)

**Comment:**

This submission on pre-training interpretable GNNs for generalizable graph interpretability has resonated positively across the board, with reviewers cohesively emphasizing its novelty, technical depth, and potential influence on the field.

* The paper's core theme of studying the pre-training of interpretable GNNs, especially the introduction of the \pi-GNN model, is perceived positively.

* The significant improvements across datasets for both interpretation and prediction aspects, especially the interpretative performance on node classification after pre-training on graph classification, strongly evince the model's cross-task generalizability. The inclusion of ablation studies, hyper-parameter analyses, and an array of visual explanation cases enriches the manuscript.

Given the unanimous appreciation for the paper's novelty, technical outline, and empirical performance, this submission is suitable for publication at NeurIPS.